# Kojic Acid Gene Clusters and the Transcriptional Activation Mechanism of *Aspergillus flavus* KojR on Expression of Clustered Genes

**DOI:** 10.3390/jof9020259

**Published:** 2023-02-15

**Authors:** Perng-Kuang Chang, Leslie L. Scharfenstein, Noreen Mahoney, Qing Kong

**Affiliations:** 1Southern Regional Research Center, Agricultural Research Service, U. S. Department of Agriculture, 1100 Allen Toussaint Boulevard, New Orleans, LA 70124, USA; 2Western Regional Research Center, Agricultural Research Service, U. S. Department of Agriculture, 800 Buchanan Street, Albany, CA 94710, USA; 3School of Food Science and Engineering, Ocean University of China, Qingdao 266003, China

**Keywords:** kojic acid, gene cluster, phylogeny, zinc cluster domain, CRISPR, *Aspergillus*, *Penicillium*

## Abstract

Kojic acid (KA) is a fungal metabolite and has a variety of applications in the cosmetics and food industries. *Aspergillus oryzae* is a well-known producer of KA, and its KA biosynthesis gene cluster has been identified. In this study, we showed that nearly all section *Flavi* aspergilli except for *A. avenaceus* had complete KA gene clusters, and only one *Penicillium* species, *P. nordicum*, contained a partial KA gene cluster. Phylogenetic inference based on KA gene cluster sequences consistently grouped section *Flavi* aspergilli into clades as prior studies. The Zn(II)_2_Cys_6_ zinc cluster regulator KojR transcriptionally activated clustered genes of *kojA* and *kojT* in *Aspergillus flavus*. This was evidenced by the time-course expression of both genes in *kojR*-overexpressing strains whose *kojR* expression was driven by a heterologous *Aspergillus nidulans gpdA* promoter or a homologous *A. flavus gpiA* promoter. Using sequences from the *kojA* and *kojT* promoter regions of section *Flavi* aspergilli for motif analyses, we identified a consensus KojR-binding motif to be an 11-bp palindromic sequence of 5′-CGRCTWAGYCG-3′ (R = A/G, W = A/T, Y = C/T). A CRISPR/Cas9-mediated gene-targeting technique showed that the motif sequence, 5′-CGACTTTGCCG-3′, in the *kojA* promoter was critical for KA biosynthesis in *A. flavus*. Our findings may facilitate strain improvement and benefit future kojic acid production.

## 1. Introduction

Kojic acid (KA) is a secondary metabolite produced by fungi, including many species of *Aspergillus* and *Penicillium* [1]. It was originally isolated by Japanese microbiologist Kendo Saito in 1907 from “koji” starter inoculum, i.e., mycelia of *Aspergillus oryzae* grown on steamed rice used in oriental food fermentation [2,3]. The chemical structure was later determined to be 5-hydroxy-2-hydroxymethyl-4-pyrone [4]. KA is a well-known inhibitor of tyrosinase [5] and its mode of action is to chelate copper ions required for tyrosinase’s activity during melanin synthesis. Hence, it has long been used as a skin-whitening agent. KA and its derivatives also possess anti-bacterial, anti-fungal, anti-inflammatory, and antineoplastic properties [6]. In the food industry, KA is used as an antioxidant to prevent enzymatic browning and extend the shelf life of fruits and vegetables. Therefore, KA has wide applications in the cosmetics, pharmaceutical, and food industries [7].

KA has been on the market for over sixty-five years. Charles Pfizer and Company, Brooklyn, New York, USA, was the first company to announce its attempt to manufacture this product [8]. Though in the early years, demands for KA were low, in recent years, rapid growth has occurred in various industries, particularly in the cosmetic sector. The global market size of KA for different applications (cosmetics, food additives, and medicine materials) in 2021 was 35 million and is expected to reach 38 million in 2028 [9]. Efforts to increase KA production by fungi include optimization of medium compositions (C, N, and C/N ratios) or culturing conditions (temperature, pH, aeration, fermentation types), or both [10]. Various carbon sources from simple C2-C7 carbohydrates to starch of agricultural staples and agro-industrial by-products can all induce KA formation to different extents [8,11]. Strain development by UV and gamma-ray radiation or mutagen treatments also has been carried out extensively [12,13].

Despite being a secondary metabolite with an all-carbon backbone, KA is quite different from those in the polyketide category. Forming units and precursors for the biosynthesis of polyketides are breakdowns of primary metabolites, such as acetates. ^14^C-labeled isotope tracer studies have proven that KA is directly synthesized from glucose. A biosynthetic scheme for KA formation, which requires at most two or three enzymes, was then proposed [14]. In a later study, experimental evidence implied the involvement of glucose dehydrogenase and gluconate dehydrogenase in KA biosynthesis. This implication was inferred from a correlation of enzyme activity patterns under various conditions with amounts of KA produced. A biosynthetic pathway based on three alternative routes was thus suggested [15]. However, up until now, the enzymes or intermediates involved have not been conclusively identified. The precise nature of the KA biosynthetic process remains virtually unclear [12]. With the advent of the genomics era, gene function studies have identified three clustered genes, *kojA-kojR-kojT*, in the *A. oryzae* genome that are responsible for KA biosynthesis. These genes encode an oxidoreductase, a transcriptional activator, and a transporter, respectively [16,17].

In this study, we surveyed the presence of KA gene clusters in *Aspergillus* and *Penicillium* species, examined their gene organization structures, and delineated phylogenetic relationships among *Aspergillus* section *Flavi* species based on KA gene cluster sequences. We then explored the transactivation mechanism of the *Aspergillus flavus* KojR regulator on the expression of *kojA* and *kojT* using a heterologous *gpdA* promoter and a homologous *gpiA* promoter to drive *kojR* expression. Lastly, we identified a consensus KojR-binding motif in the promoter regions of section *Flavi* aspergilli to be an 11-bp palindromic sequence of 5′-CGRCTWAGYCG-3′ (R = A/G, W = A/T, Y = C/T). Using a CRISPR/Cas9-mediated gene-targeting approach, we showed that the motif sequence, 5′-CGACTTTGCCG-3′, in the *A. flavus kojA* promoter was critical for KA biosynthesis.

## 2. Materials and Methods

### 2.1. Media and Culture Conditions

For spore production, *A*. *flavus* was grown on V8 agar medium, which contains 5% V8 vegetable juice (Campbell Soup Company, Camden, NJ, USA) and 2% agar, pH 5.2. Kojic acid medium (KAM) for KA production [16] was modified slightly to contain 0.05% yeast extract, 0.1% K_2_HPO_4_·3H_2_O, 0.05% MgSO_4_·7H_2_O, and 2% glucose (pH 6.0). A potato dextrose agar medium (Difco, Detroit, MI, USA) supplemented with 1 mM ferric ion also was used for the examination of KA production. All fungal cultures were grown at 30 °C in the dark.

### 2.2. Identification of KA Gene Clusters in Aspergillus Species

The genome sequence of *A. flavus* NRRL3357 was retrieved from the National Center for Biotechnology Information (NCBI) genome database (https://www.ncbi.nlm.nih.gov/genome/). Analyses of the genome sequence was performed via CoGe (Comparative Genomics, https://genomevolution.org/coge/), an online platform for sequence comparison and retrieval. The extracted complete KA gene cluster sequence of NRRL3357, derived from comparisons to *A. oryzae* KA gene cluster sequences (*kojA*, *kojR,* and *kojT*) (https://www.genome.jp/kegg/), was used as the alignment template for BlastN genome assemblies of other aspergilli available in NCBI (Table 1). Aligned genomic sequences were downloaded for further analysis. The genes of the *A. flavus* KA gene cluster (*kojA*, *kojR*, and *kojT*), previously known as AFLA_096040, AFLA_096050, and AFLA_096060, are now updated as AFLA_009845, AFLA_009846, and AFLA_009847.

### 2.3. Phylogenetic Study of Section Flavi Aspergilli

The online system Mauve (http://darlinglab.org/mauve/mauve.html) for multiple genome alignments [18] was used to align the KA gene cluster sequences of twenty-three *Aspergillus* species (Table 1). Aligning sequences, extracting single nucleotide polymorphisms (SNP), filtering out noise (i.e., gaps and ambiguous bases), concatenating cleaned SNPs, and converting sequences to FASTA format were carried out using a custom JavaScript [19]. Phylogenetic inference with concatenated total SNP sequences was performed using the unweighted distance-matrix UPGMA method of the online program MAFFT (version 7) (https://mafft.cbrc.jp/alignment/server/phylogeny.html) [20].

### 2.4. Disruption of the kojR Gene in A. flavus

Gene deletion procedures via double-crossover recombination in *A. flavus* strains deficient in the NHEJ pathway have been described in detail [21]. This well-established protocol is routinely used for gene function studies by many researchers to obtain gene knockout mutants. The two 5′ and 3′ flanking fragments for targeting *kojR* were amplified by PCR using AccuPrime^TM^ Pfx Taq polymerase (Invitrogen, Carlsbad, CA, USA) with primers listed in Appendix A. The two PCR fragments, after digestion with appropriate restriction enzymes, were sequentially cloned into a *pyrG*-containing plasmid, pPG28. The resulting deletion vector was linearized with *Sac*I and *Sph*I prior to polyethylene glycol-CaCl_2_ mediated fungal transformation [21]. The recipient strain, *A. flavus* SRRC1709 (=CA14PTS∆pyrG), was pyrithiamine-sensitive and uracil auxotrophic. All *kojR* deletion mutants were confirmed by diagnostic PCR and verified for their inability to produce KA. A deletion mutant, ∆kojR#4, was used for subsequent genetic complementation experiments.

### 2.5. Construction of kojR Expression Vectors with gpdA or gpiA Promoter

The pTR1-GPD-TRPC vector that contains the *A. nidulans* glyceraldehyde-3-phosphate dehydrogenase gene (*gpdA*, AN8041) promoter and *trpC* terminator and the *A. oryzae* pyrithiamine resistance gene (*ptrA*) as the selection marker [22] was modified slightly. Transcriptomic information based on RNA-Seq reads available from now-delisted AspGD (http://www.aspergillusgenome.org/) revealed that only the upstream 1.0 kb (579170 to 580140) is the *gpdA* promoter region, which is consistent with the previous report [23]. Therefore, the 1.1-kb upstream region in the vector that is not a part of the *gpdA* promoter but contains the phosphate transporter gene, AN8040, was removed by *Hin*dIII and *Xho*I digestion followed by self-ligation. This derived cloning vector was named pHgpdA-trpC-PTR in this study. A full-length *A. flavus* genomic *kojR* coding sequence was PCR amplified with primers kojR-OE and kojR-STOP (Appendix A) by AccuPrime^TM^ Pfx polymerase and cloned into the *Not*I and *Sma*I sites of the shortened vector.

*A. flavus gpiA* (previously named AFLA_113120 and now as AFLA_007250 in FungiDB, https://fungidb.org/fungidb/app/), an orthologue of *Saccharomyces cerevisiae* ecm33, is a highly expressed gene [24,25]. An analysis of RNA-Seq reads related to this gene indicated that a 1.4-kb region, including the 5′ UTR, contained its promoter. A PCR fragment of the specified region was generated with primers GPI-H and GPI-Not (Appendix A) to replace the *gpdA* promoter of pHgpdA-trpC-PTR to give pHgpiA-trpC-PTR. The vector had a unique *Not*I site and a unique *Sma*I site. Lastly, the full-length *A. flavus kojR* coding sequence was PCR amplified and cloned. The complementation vectors were individually transformed into the ∆kojR#4 mutant. Three KA-producing transformants from the *A. nidulans gpdA* promoter set (D-8, D-16, and D-20) and the *A. flavus gpiA* promoter set (I-5, I-9, and I-16) were selected based on visual inspection showing high levels of KA production (Appendix A). Their *kojR* copy numbers were determined, and they were used in a time-course expression study to examine how the restoration of *kojR* expression affected the expression of *kojA* and *kojT*.

### 2.6. Determination of KA Production and Fungal Mycelial Dry Weight

Spores of the six *kojR*-overexpressing strains and a control strain, KuPG, were harvested from five-day-old V8 agar plates, washed to remove debris, and counted on a hemocytometer. They were inoculated into 50 mL of KAM broth in 150 mL Erlenmeyer flasks at a final concentration of 10^5^ per mL. Cultures in triplicate were incubated in a gyroshaker at 30 °C and shaken at 150 rpm for four days. At the end of growth, 10 mL aliquots of culture broth were collected and frozen at −20 °C. Mycelia were spun down, supernatants decanted, and transferred to 15-mL polypropylene tubes for lyophilization. After completion, mycelial dry weights were determined. For HPLC analysis of KA, frozen filtrates were thawed at room temperature, and a 1.5 mL aliquot of each was filtered through a 25 mm 0.45 µm nylon syringe filter. Each filtrate was diluted one to twenty times with water and a 20 µL portion was analyzed using a column of Intersil ODS-3, 5 μ, 4.6 × 250 mm (GL Sciences, Torrance, CA, USA) at a flow rate of 1.0 mL/min. The mobile phase was MeOH:0.1% H_3_PO_4_ (25:75), isocratic. The detection of KA was based on UV absorption at 265 nm. A linear standard curve constructed from standard KA concentrations was used to determine the amounts of KA in the filtrate samples.

### 2.7. Determination of kojR Copy Numbers of gpdA and gpiA Promoter-Driven Overexpression Transformants

Spores of 1.0 × 10^4^ of the *kojR-*overexpressing and the KuPG control strains, prepared from V8 medium plates, were inoculated, respectively, into 1.0 mL PDB in 2-mL microfuge tubes. The cultures were grown stationary at 30 °C for two days. Mycelia were then harvested and disrupted with ZR bashing beads using a Mini-Beadbeater-8 cell disrupter (Biospec Products, Bartlesville, OK, USA). Genomic DNA was purified using a ZR Fungal/Bacterial DNA MiniPrep kit (Zymo Research, Orange County, CA, USA). The concentration of genomic DNA was determined with a Thermo Scientific™ NanoDrop™ 2000 spectrophotometer. qPCR was used to determine the copy numbers of overexpression strains using 20 ng of DNA as templates. Three technical replicates were used for each sample. The primers were kojR-F: AATACCGACGATTCCGGTCG and kojR-R: TTTCCTCTTGCGCAGTTTGC.

### 2.8. Time-Course Quantitative Reverse Transcription PCR (qRT-PCR) Analysis

Cultures for qRT-PCR analyses were grown on duplicate Petri dishes (100 × 15 mm), each containing 25 mL PDB and 1.0 × 10^6^ spores. At 48 and 72 h, mycelia were harvested, rinsed with sterilized distilled water, and ground with a mortar and pestle under liquid nitrogen. Total RNA was extracted with a Direct-Zol RNA Miniprep Kit (Zymo Research, Irvine, CA, USA) and followed by DNase I treatment. qRT-PCR was carried out in a 20 μL reaction volume with the LuminoCT SYBR Green qPCR ReadyMix (Sigma-Aldrich, St. Louis, MO, USA) and the reverse transcriptase enzyme mix (Applied Biosystems, Foster City, CA, USA) in an Applied Biosystems StepOne^TM^ thermal cycler. The amplification conditions were as follows: an initial step of 48 °C for 30 min for reverse transcription reactions was followed by forty cycles, each consisting of 95 °C for 5 s and 60 °C for 20 s. Four technical replicates were used for each sample. Gene-specific primers used were listed in Appendix A. Relative expression levels were calculated using the 2^−ΔΔCt^ method.

### 2.9. KojR DNA-Binding Motif Analysis

The motif finding program Multiple EM for Motif Elicitation (MEME, Version 5.3.0, http://meme-suite.org/tools/meme) [26] was used to predict the putative KojR-binding motif sequences in both *kojA* and *kojT* promoters of the twenty-three aspergilli in the section *Flavi*. Sequences of around 370 and 380 nucleotides in the respective 5’-untranslated regions of *kojA* and *kojT* were extracted from genome assemblies (Appendix A) and uploaded into MEME for motif analysis.

### 2.10. Generation of Defects in Putative KojR DNA-binding Sites by the Established CRISPR/Cas9 Genome-Editing Approach

Targeted alteration of putative KojR DNA-binding sites was carried out using the established genome-editing CRISPR/Cas9 technology [27]. pAf-CRISPR-yA (Addgene plasmid #191015) was the template for making DNA fragments that encoded single guide RNAs (sgRNAs) to target motif sequences. The PCR fragments were cloned into the *Pst*I and *Kpn*I sites of pAsp-AMA-gpdA-ptr (Addgene plasmid #191016) to give the resulting gene-targeting vectors. The target sequences in *kojA* and *kojT* for the CRISPR/Cas9 complexes were GGTGGAATGAGCGGCAAAGTCGG and AAGCCATTCAGCGGCTAAGTCGG, respectively (motif sequences are underlined). To construct a sgRNA expression cassette, two target-specific DNA fragments were first generated by PCR with primer sets of U6-F-P/kojA_R or kojT_R and kojA_F or kojT_F/U6-R-K, respectively (Appendix A). To this end, 40 pmol of each primer and 2 ng template were added to 20 μL of AccuPrime™ SuperMix (Invitrogen) and subjected to thirty cycles of PCR, which consisted of denaturation at 94 °C for 30 s, annealing at 55 °C for 30 s and extension at 72 °C for 1.0 min. The two PCR fragments, without further purification, were directly fused and amplified by another round of PCR using primers U6-F-P and U6-R-K. The annealing time was set for 2 min. The resulting fragment, after being cut with *Pst*I and *Kpn*I, was cloned to give a final vector. Gene-targeting vectors were transformed into the wild-type *A. flavus* CA14 recipient as previously described [21]. Selected primary pyrithiamine-resistant transformants were transferred onto KAM plates and grown at 30 °C for two to three days for direct PCR and sequencing analyses to confirm indel defects in the targeted *kojA* and *kojT* genes.

### 2.11. Identification of Molecular Defects in kojA and kojT Transformants

Sequencing was performed to reveal defects in *kojA* transformants that lost KA production and in *kojT* transformants that retained or lost KA production. Direct PCR using fungal mycelia as the genomic template and location-specific primers that encompassed targeted sequences was carried out with a Phire Plant Direct PCR Master Mix (Thermo Fisher Scientific, Waltham, MA, USA) [28]. The PCR protocol consisted of an initial denaturation at 98 °C for 5.0 min, followed by forty cycles of denaturation at 98 °C for 5 s, annealing at 60 °C for 5 s, and extension at 72 °C for 30 s. Purified PCR products were sequenced at the Genomics and Bioinformatics Research Unit of the Agricultural Research Service, US Department of Agriculture (Stoneville, MS, USA).

## 3. Results

### 3.1. Phylogeny of Aspergillus Section Flavi Species Bases on Complete KA Gene Cluster Sequences

We investigated the presence of a complete KA gene cluster, that is, *kojA-kojR-kojT* in over 100 *Aspergillus* species, whose genome sequences are publicly available at NCBI. The sequence of *the A. flavus* NRRL3357 KA gene cluster (CP044621.1: 4359976 to 4365751), which resides on chromosome five, was the alignment template. We found that twenty-three known species contained the complete KA gene cluster (Table 1). They exclusively belonged to the *Flavi* section (“section” is a taxonomic rank in-between genus and species). Phylogenetic inference based on variations in conserved sites in the KA gene clusters showed that they were grouped into six clades (Figure 1). *A. flavus*, *A. oryzae*, *A. minisclerotigenes* and *A. aflatoxiformans* (=*A. parvisclerotigenus*) were in one clade, well separated from *A. parasiticus*, *A. novoparasitcus*, *A. sojae*, *A. arachidicola* and *A. transmontanensis*, which were in another clade. The phylogeny also showed that *A. oryzae* was closer to *A. aflatoxiformans* than to S-morphotype *A. flavus*, and it was adequately separated from *A. minisclerotigenes*. All known aflatoxin producers, *A. flavus*, *A. pseudonomius*, *A. aflatoxiformans*, *A. parasiticus*, *A. novoparasiticus*, *A. arachidicola*, *A. minisclerotigenes*, *A. transmontanensis*, *A. nomiae*, *A. luteovirescens*, and *A. sergii* had complete KA gene clusters. 

### 3.2. Partial KA Gene Clusters in Other Aspergilli and Penicillia

The presence of a *kojR* homolog (An09g05060) in *Aspergillus niger*, which encodes a protein with 63% amino acid sequence identity to *A. oryzae* and *A. flavus* KojR proteins, has been reported [17]. In this study, we further confirmed that An09g05070, situated next to An09g05060 and encoding an MFS (major facilitator superfamily) transporter protein, was a *kojT* homolog with 78% amino acid sequence identity. However, we did not find homologous genes encoding the KojA oxidoreductase next to or adjacent to An09g05060. Therefore, *A. niger* contained only a partial KA gene cluster. We identified homologous proteins from other twenty-one aspergilli, including *A. avenaceus*, a section *Flavi* species (Table 2), using both annotated amino acid sequences of *A. niger* KojR and KojT as alignment templates. Surprisingly, no genes encoding a KojA oxidoreductase were present near *kojR* in *A. avenaceus* or the other twenty aspergilli not belonging to section *Flavi*. To identify possible KA gene clusters in *Penicillium* species, we carried out a BlastP search using the KojR and KojT sequences of *A. steynii* and *A. tanneri*, which had the lowest degrees of amino sequence identity to the homologs of the rest of the aspergilli. The rationale was that the low degrees may suggest that they are closer to *Penicillium* or other fungal genera than to other aspergilli. By this approach, we further identified KojR and/or KojT homologs from *P. nordicu*m, *P. freii*, and *P. polonicum*, among over thirty *Penicillium* species. Using the *P. nordicum* KojR protein sequence (KOS46106.1) as an alignment template, we obtained the same results. The *Penicillium* KojR and KojT amino acid sequences were 50–56% and 70–73% identical to those of the homologs of *A. steynii* and *A. tanneri*, respectively). However, many of the originally annotated *Penicillium* KojR proteins varied greatly in their size, and some were mis-annotated. We found that only *kojR* and *kojT* of *P. nordicu*m were organized as a partial gene cluster (Table 2). 

### 3.3. Expressing kojR Restored Expression of kojA and kojT in Overexpression Strains

The *A. flavus kojR* gene was involved in KA production, and *kojR* knockout strains were unable to produce KA. To investigate the transcriptional activation mechanism for *kojA* and *kojT* expression, we fused *kojR* with the *gpdA* or *gpiA* promoter and carried out overexpression experiments. Transformants exhibited varied abilities in terms of KA production on KAM or PDA supplemented with ferric ions (Appendix A). We selected three putative overexpression strains that visually showed high amounts of KA production from each set and determined their *kojR* copy numbers. Only one strain (D-8) in the *gpdA* promoter set contained more than one copy of *kojR*. In contrast, the strains of the *gpiA* set contained two to three copies of *kojR* (Figure 2A). In the time-course study that examined how *kojR* expression affected the expression of *kojA* and *kojT*, we found that the ∆kojR#4 mutant had lost *kojR* expression; its normalized expression levels of *kojA* and *kojT* at 72 h compared with those at 48 h decreased slightly and were within 40% (Appendix A). For the KuPG control strain, normalized expression levels of *kojA* and *kojT* from 48 h to 72 h increased 2.0- and 35.2-fold, respectively. Noticeably, the *kojR* expression levels of the control and overexpression strains did not change much from 48 h to 72 h, and the variations were within 70%. Both the *gpdA* and *gpiA* promoters were able to drive higher *kojR* expression in the overexpression strains than the native *kojR* promoter in the control strain. We observed that *kojR* expression levels in all overexpression strains compared to those in the control strain were higher at 48 h than at 72 h. The relative expression levels at 48 h somewhat reflected the *kojR* copy numbers of these strains (Figure 2B). The normalized *kojA* and *kojT* expression levels in all overexpression strains at 48 h were low but increased significantly at 72 h (Appendix A). Using the normalized *kojA* and *kojT* expression levels of the ∆kojR#4 mutant, which did not vary much from 48 h to 72 h (Appendix A), as the basal expression levels (relative fold treated as one), we found that the relative *kojA* and *kojT* expression levels in most of the strains were low at 48 h except for D8, which were comparable to the control levels (Figure 2C). Noticeably, the *kojA* and *kojT* expression levels in D-16 and D-20 were significantly lower than those in the ΔkojR strain. These decreases seem to be related to delayed gene expression at 48 h in the two heterologous *gpdA* promoter set strains, which contained a single copy of *kojR* (Figure 2A), as compared to the *kojA* and *kojT* expression in the homologous *gpiA* promoter set strains at 48 h (Appendix A). It is not known whether the developmental or physiological states of the two strains have a bearing on the observed variation. Expression levels of *kojA* and *kojT* at 72 h increased substantially from the basal levels but also varied greatly. Only the D8 relative expression patterns resembled those of the control (Figure 2C). Table 3 shows that at 96 h only strain D-16 produced 43% more KA than the control strain, whereas others produced slightly lower or higher amounts of KA.

### 3.4. Zn(II)_2_Cys_6_ Zinc Cluster Domains and Downstream Basic Regions of Aspergilli

We analyzed the amino-terminal 120 amino acid-containing regions of KojR proteins of the twenty-three-section *Flavi* aspergilli. Each region included a binuclear zinc-finger DNA-binding domain and the downstream basic sequence associated with dimerization. Using *A. flavus* KojR as a reference, we observed amino acid variations, including deletions and conservative and radical substitutions, in some aspergilli. All twenty-three aspergilli had identical Zn(II)_2_Cys_6_ zinc cluster domains, **C**ET**C**KLRKRK**C**DGHEP**C**TY**C**LRYEYQ**C** (Table 4). For the two aspergilli not belonging to section *Flavi*, *A. phoenicis* and *A. welwitschiae* (Table 2), they had the same Zn(II)_2_Cys_6_ zinc-cluster as *A. flavus* KojR. However, others showed minor variations. For example, the corresponding domain in *A. niger* was CETCKLRKRKCDGHEPC$\underset{̿}{SF}$CL$\underset{̿}{K}$YEY$\underset{̿}{D}$C and contained four amino acid substitutions.

### 3.5. Identification of Putative KojR-Binding Sites in kojA and kojR Promoters of Aspergilli

The alignment of zinc-cluster domains in section *Flavi* aspergilli suggested that they recognized the same binding motif(s). To identify possible consensus KojR-binding site(s), we examined sequences in the *kojA* and *kojT* promoter regions of twenty-four aspergilli, including *A. flavus* L- and S-morphotype. Table 4 shows that the *kojA* and *kojT* promoter sequences of *A. flavus* shared various degrees of nucleotide sequence identity with those of the other twenty-two aspergilli, which ranged from 63.9% to 100% and 60.5% to 99.5%, respectively. We next performed motif analysis using MEME on sequences from half of the intergenic regions of *kojA* and *kojR* (370 bps upstream of *kojA*) and in the complete inter-gene regions of *kojR* and *kojT* (383 bps) (Appendix A). We obtained a putative DNA-binding motif that contained an 11-nucleotide palindromic sequence, 5′-CGRCTWAGYCG-3′ (R = A/G, W = A/T, Y = C/T) (Figure 3). All putative motif sequences found in the *kojA* and *kojT* promoters were situated within 0.3 kb of the ATG start codon. Table 5 shows that these putative motif sequences (from 5′ to 3′) are on positive strands in the *kojA* promoters. However, only reverse complementary sequences that are like the aforementioned motif sequences (with one nucleotide substitution, i.e., position seven changed from T to A) are present on the positive strands in the *kojT* promoters (relative to the ATG codon). In other words, putative motif sequences were found on both strands when viewed because of their palindromic nature.

### 3.6. Involvement of the KojR-Binding Site in the A. flavus kojA Promoter in KA Production

To corroborate the significance of the identified motif sequences in the *kojA* and *kojT* promoters in KA biosynthesis, we carried out site mutation analyses in *A. flavus* using a CRISPR/Cas9-based genome editing technique. This technology is well-known for generating small deletions in targeted sequences. We analyzed twenty and thirty-four primary transformants of the *kojA* and *kojT* sets, respectively, for KA production by transferring them onto KAM plates. For the *kojA* set, no transformants were produced for KA. For the *kojT* set, twenty-three produced KA but eleven did not. PCR examination of the fifty-four transformants with location-specific primers (Appendix A) revealed that seven transformants from the *kojA* set and seventeen transformants from the *kojT* set (fifteen KA producers and two KA non-producers) gave positive PCR products. Subsequent sequencing of these PCR products showed that the KA-nonproducing *kojA* transformants contained six types of sequence defects, which consisted of five types of deletions and one type of insertion. These defects disrupted various parts of the putative KojR-binding motif in the *kojA* promoter (Figure 4A and Appendix A). Similarly, KA-producing *kojT* transformants had nine types of defects, which consisted of six types of deletions and three types of insertions (one, six, and one hundred five nucleotides) that disrupted the putative KojR-binding motif in the *kojT* promoter. Figure 4B shows six types of them. The transformant T3 had half of the motif plus the downstream region extending beyond the *kojT* start site (404 bps) deleted (Appendix A). The two *kojT* transformants, T18 and T21, that did not produce KA, had large deletions of 999 and 1566 nucleotides, respectively. This started at the *kojT* target site and extended to the upstream *kojR* coding region (Appendix A), which indicated that the loss of KA production resulted primarily from the defective *kojR*. Figure 4C depicts the location of the identified functional KojR-binding site in the KA gene cluster. A defect at this site resulted in the loss of KA production.

## 4. Discussion

A list of thirty-three accepted species and their synonyms in *Aspergillus* section *Flavi* has been published [29]. Most recently, an additional species, *A. burnettii*, was identified [30]. Our analyses of the genome sequences of more than one hundred *Aspergillus* species showed that the KA gene cluster, *kojA*-*kojR*-*kojT*, was present nearly exclusively in *Flavi* section members (Table 1). Therefore, they appear to be the only aspergilli capable of producing KA. Phylogeny inferred from the KA gene cluster sequences showed that the derived clades were like those based on multi-locus sequences or 200 monocore (i.e., single copy) gene sequences [29,31]. The *Aspergillus sp.* ATCC 12892, officially designated as *Aspergillus oryzae* (Ahlburg) Cohn (ATCC^®^ 12892™), turned out to be misidentified. It is most likely a strain of *A. parasiticus* or a closely related species. Its genome size (41.2 kb), which is 10% larger than those of *A. oryzae* RIB40 and *A. flavus*, also supports this notion. Misclassification of *Apergillus* species has been a concern, especially in the genomic era [32]. Classification of *Aspergillus* species by multi-locus sequence typing has been a conventional approach. The current study shows that using SNPs in the KA gene clusters is an equally valid method, and it can distinguish a misidentified species from correctly named aspergilli.

KA production by most sections *Flavi* aspergilli has been reported. However, not all *Flavi* species can produce KA, for example, *A. coremiiformis* and *A. avenaceus* [10,29]. The genome size of *A. coremiiformis* was 20% smaller than the average (Table 1). Nonetheless, it had a complete KA gene cluster. This KA gene cluster sequence, however, did not have a significant nucleotide identity to that of *A. flavus*. Cross-species sequence comparisons further showed that it shared 80–82% nucleotide identity with those of *A. alliiaceus* and *A. burnettii* in the same clade (Table 1 and Figure 1). We noticed that sequences in the promoter regions of *A. coremiiformis kojR, kojA*, and *kojT* were also highly variable (Appendix A), which may explain its inability to produce KA. The *A. oryzae* KA gene cluster (*kojA*-*kojR*-*kojT*) identified in 2010 is likely an incomplete gene cluster. More than one enzyme is required for KA formation [15]. The *kojA* gene product, an oxidoreductase, should be directly involved in the conversion of the glucose precursor or intermediate metabolites. However, neither *kojR* nor *kojT*, which encode a Zn(II)_2_Cys_6_ zinc cluster regulator and a transporter protein (efflux pump), respectively, possess enzymatic activities. Therefore, we cannot exclude the possibility that the inability of *A. coremiiformis* to produce KA is due to defects in another yet-to-be-identified gene(s). 

*A. niger* has been reported as being unable to produce KA [1]. In this study, we found that it only contained a partial KA gene cluster. Hence, the inability of *A. avenaceus* to produce KA is probably due to not having a complete KA gene cluster. Other aspergilli not belonging to section *Flavi* but containing partial KA gene clusters (Table 2) likely do not produce KA either. Up to now, only a few *Penicillium* species, such as *P. citrinum*, *P. griseofulvum*, *P. purpurogenum*, *P. rubrum*, *P. jamesonlandense,* and *P. lanosum* are known to produce KA [1,33]. Of the six species, the genome sequences of *P. citrinum* and *P. griseofulvum* are available in the NCBI genome database. Nonetheless, neither of the two penicillia contained a complete KA gene cluster. Our finding that only *P. nordicum* had a partial KA gene cluster suggests that most *Penicillium* KA biosynthesis genes, if they exist, do not form a gene cluster.

Zn(II)_2_Cys_6_ type regulators are unique to fungi and yeast [34,35,36]. They are involved in a diverse array of functions [37]. In the *A. oryzae kojR*-overexpressing strain, expression of *kojA* and *kojT* was barely detected at 24 h until 72 h, when the levels increased substantially [17]. In this study, the six *A. flavus kojR*-overexpressing strains also produced relatively low levels of *kojA* and *kojT* transcripts at 48 h, but the levels were greatly elevated at 72 h. Through transcriptional activation of *kojA* and *kojT* expression driven by the heterologous *A. nidulans gpdA* promoter and the homologous *A. flavus gpiA* promoter KA production was restored (Figure 2 and Table 3). However, it is not known if autoregulation, that is, a direct modulation of gene expression by the product of the corresponding gene (KojR) to activate or suppress its own transcription [38,39], is involved in *kojR* expression in wild-type *A. flavus*. The extents of variation in normalized *kojR* expression from 48 h to 72 h in the control and overexpression strains were much lower than those in *kojA* and *kojT* (Appendix A). This implies that the pathway-specific *kojR* is tightly regulated, probably by unidentified global regulators. In the *A. oryzae* study [17], KA was reported to increase the transcription of its biosynthesis genes. Although *kojA* and *kojR* share the same intergenic region, the effect of KA addition was more prominent on *kojA* than on *kojR*. Feedback regulation, the process by which a metabolic pathway product influences its own production by controlling the amount and/or the activity of one or more involved enzymes [40], may have a bearing on KA biosynthesis.

A consensus motif sequence, 5′-AGTCGGG-3′ in upstream regions of *A. oryzae kojA* and *kojT* was proposed to be responsible for the regulation of *kojA* and *kojT* [17]. However, our analysis of the *A. oryzae* KA gene cluster showed that this heptanucleotide sequence resided on different strands, that is, on the negative strand upstream of *kojA* but on the positive strand upstream of *kojT* (Table 5). The difference in motif sequence orientation suggests that *kojA* and *kojT* might not be coordinately regulated by KojR. The proposed *A. oryzae* sequence (bold) was a part of 5′-CGGCAA**AGTCG****GG**-3′on the positive strand upstream of *kojR* (underlined is the reverse complementary sequence of 5′-CGACTTTGCCG-3′, the motif identified in the *A. flavus kojA* promoter). Zn(II)_2_Cys_6_ type regulators like KojR bind as a dimer to both DNA strands [37]. The physical distance from the motif site to the *kojR* coding sequence start codon, ATG, was approximately twice that from the motif site to the *kojA* start codon (Table 5 and Figure 4C), which suggests that KojR is likely mainly involved in *kojA* regulation.

The CRISPR/Cas9 technology could be used as a complementary method to the ChIP-seq method for identifying DNA-binding motifs or the electrophoretic mobility shift assay (EMSA) for detecting protein-DNA interactions [41,42]. In this study, we showed that only the 11-bp sequence motif in the *A. flavus kojA* promoter was required for KA production (Figure 4A). Hence, the genuine KojR-binding motif sequence for aspergilli is likely longer than what was proposed for *A. oryzae*. For example, both the 18-bp insertion in the transformant A7 and the 8-bp deletion in the transformant A15 did not disrupt the *A. flavus* equivalent of the *A. oryzae* motif 5′-**CCCGACT**TTGCCG-3′ (bold is the reverse complementary sequence of 5′-AGTCGGG-3′) in their *kojA* promoters (Figure 4A). It was the deletions upstream of 5′-AGTCGGG-3′ that rendered the two *A. flavus* mutants unable to produce KA. For the *A. flavus* promoter, disruptions of the 11-bp palindromic motif 5′-CGGCTAAGTCG-3′ in transformants T2, T7, and T11 or a complete deletion of the proposed *A. oryzae* motif sequence 5′-AGTCGGG-3′ in transformants T3, T24, and T32 did not abolish KA production (Figure 4B). However, in the characterization of *A. oryzae kojT* by Terabayashi et al. [16], the *kojT* deletion mutant still produced a reduced amount of KA, as evidenced by the colony showing a pale red color. Therefore, we cannot rule out that KA production in these *A. flavus* mutants results from the presence of another unidentified transporter protein that can replace the function of KojT, as previously proposed. Nevertheless, the present work shows that the motif in the *A. flavus kojA* promoter is highly likely a KojR-binding site and is critical for KA production. Zn(II)_2_Cys_6_-type proteins act as pathway-specific regulators [37]. Three, all named AflR, are involved in the biosynthesis of sterigmatocystin and aflatoxin in *A. nidulans* and *A. flavus/A. parasiticus*, respectively, have been characterized; and their binding motifs are 11-bp palindromic sequences [43,44,45]. Apparently, *A. flavus* KojR has the same binding capacity. Despite the initial characterization of KojR-binding sites in the present study, the final proof of the identity of the KojR-binding motif, including its actual length, will have to come from direct evidence of protein-DNA interaction, such as the use of ChIP technology. Future characterization of those yet-to-be-identified structural genes that are directly involved in KA formation will shed light on genuine KojR-binding sequences in the KA biosynthesis gene promoters.

## Figures and Tables

**Figure 1 jof-09-00259-f001:**
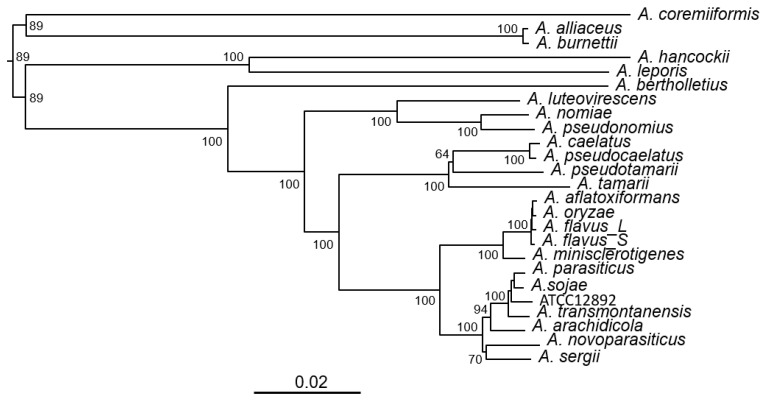
Phylogenetic tree of twenty-four *Aspergillus* section *Flavi* species inferred from KA gene cluster sequences using NJ analysis. A total of 5776 conserved sites (i.e., concatenated sequences of total SNPs) from each species were used. Bootstrap values are shown at the nodes. The branch length scale is shown. Branch lengths represent genetic change; the longer the branch, the more divergence has occurred. The exact species name of ATCC12892, originally designated as *A. oryzae*, is not known.

**Figure 2 jof-09-00259-f002:**
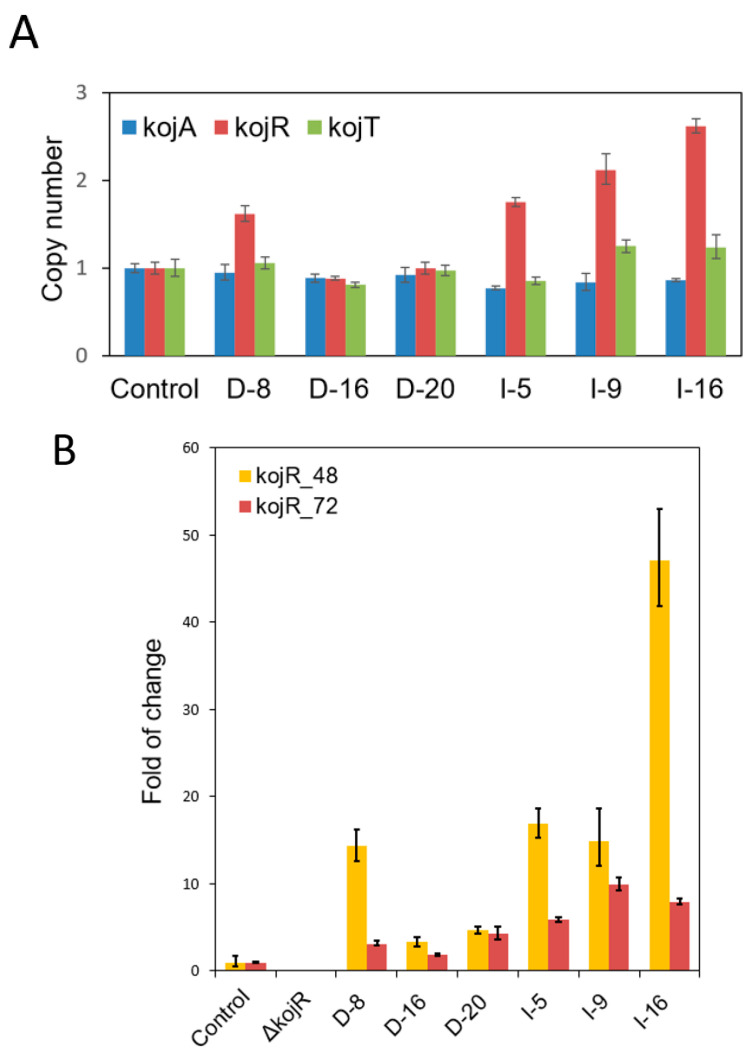
Determination of *kojR* copy numbers and relative expression levels of *kojR*, *kojA*, and *kojT* of *kojR*-overexpressing strains. (**A**) *kojR* copy numbers of overexpression strains whose *kojR* expression was driven by the *A. nidulans gpdA* promoter or the *A. flavus gpiA* promoter. The copy numbers of *kojA* and *kojT* of the control and overexpression strains were used as single-gene-copy checks. (**B**) Relative expression levels of *kojR* to those of the control strain at 48 h and 72 h. (**C**) Relative expression levels of *kojA* and *kojT* to those of the ∆kojR strain, which presumably were the basal expression levels at 48 h and 72 h.

**Figure 3 jof-09-00259-f003:**
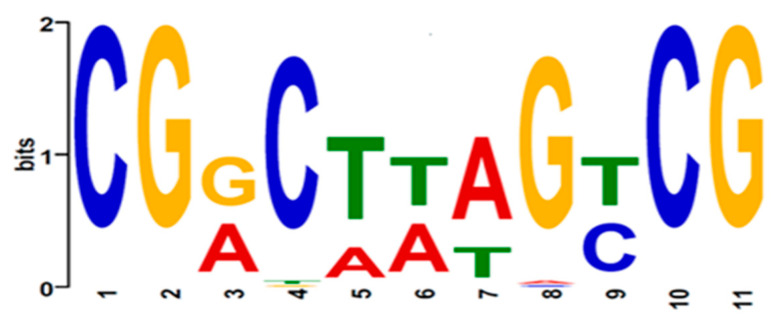
Putative KojR-binding motif identified by MEME using *kojA* and *kojT* promoter sequences of section *Flavi* aspergilli. The logo is the downloaded EPS (for publication) version from the MEME site, whose appearance is somewhat different from the PNG (for web) version in that all positions have the same baseline. The relative height indicates how certain it is to observe a particular nucleotide at a particular position, and high heights indicate high conservation/low uncertainty. In the MEME analysis, the maximum motif width was arbitrarily set at eleven and searched for palindromic motifs. Promoter sequences listed in Appendix A are the input sequences.

**Figure 4 jof-09-00259-f004:**
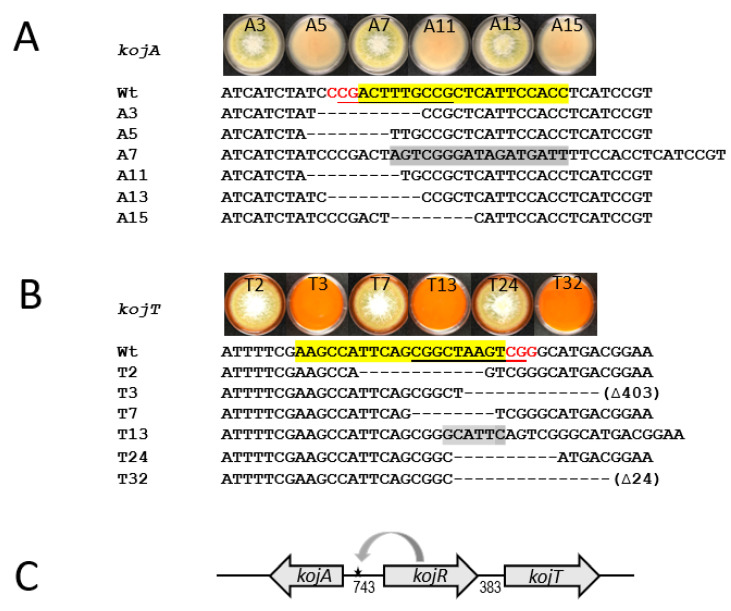
Identification of KojR-binding site in *A. flavus kojA* promoter. (**A**) Six indel defects in the motif of *kojA*, 5′- CGACTTTGCCG-3′, rendered the transformants unable to produce KA. (**B**) Six indel defects disrupted the identified motif of *kojT*, 5′-CGGCTAAGTCG-3′. However, they did not affect KA production of the transformants. The recipient strain used for the CRISPR/Cas9 work is wild-type *A. flavus* CA14. Wt represents wild-type sequences. Yellow-highlighted sequences are the target sites of the CRISPR/Cas9 complexes. Red trinucleotides CCG and CGG are protospacer adjacent motifs (PAM) that follow the regions targeted for cleavage by the Cas9 nuclease. Dash lines are deleted sequences. Gray-highlighted sequences are additional nucleotides inserted into respective motifs. The symbol ∆403 indicates a large deletion extending to the *kojT*-coding sequence (see Appendix A). The photos above the sequences are colony morphologies of six mutants on KAM agar plates, which are shown alternatively on their front and reverse sides. Colonies were grown at 30 °C for five days in the dark. Orange-red plates are KA-producing colonies. (**C**) Graphic representation showing the location of the functional KojR-binding site in the *kojA* and *kojR* intergenic regions inferred from the present study. The site is 266 nucleotides from the translation start codon of *kojA* and 466 nucleotides from the start codon of *kojR*.

**Table 1 jof-09-00259-t001:** Genome sequences of *Aspergillus* section *Flavi* species and their KA gene clusters.

Species	Strain	Genome (Mb)	GenBank (WGS)	Identity (%) ^a^	Note
*A. aflatoxiformans*	CBS 121.62	37.6	SWAT00000000.1	99.8	=*A. parvisclerotigenus*
*A. alliaceus*	CBS 536.65	40.2	SWAS00000000.1	81.6	=*A. albertensis*
*A. arachidicola*	CBS 117610	38.9	NEXV00000000.1	96.5	
*A. avenaceus*	IBT 18842	33.8	STFI00000000.1	Not significant	
*A. bertholletius*	IBT 29228	37.0	STFP00000000.1	87.2	
*A. burnettii*	FRR 5400	41.0	SPNV00000000.1	81.6	
*A. caelatus*	CBS 763.97	40.0	STFO00000000.1	92.5	
*A. coremiiformis*	CBS 553.77	30.1	STFN00000000.1	78.3	80.09% to *A. leporis* 81.53% to *A. alliiaceus* 81.53% to *A. burnettii*
*A. flavus*	NRRL 3357	36.9	AAIH00000000.3	100.0	L-morphotype
*A. flavus*	AF12	38.0	NLCN00000000.1	99.8	S-morphotype
*A. hancockii*	FRR 3425	39.9	MBFL00000000.1	80.4	
*A. leporis*	CBS 151.66	39.4	SWBU00000000.1	80.9	
*A. luteovirescens*	NRRL 26010	37.5	LYCR00000000.1	91.6	=*A. bombycis*
*A. minisclerotigenes*	CBS 117635	37.1	SWDZ00000000.1	98.9	
*A. nomiae*	NRRL 13137	36.1	JNOM00000000.1	91.2	=*A. nomius*
*A. novoparasiticus*	CBS 126849	40.9	SWDA00000000.1	96.3	
*A. oryzae*	RIB40	37.1	JZJM00000000.1	99.8	
*A. parasiticus*	SU-1	39.5	JZEE00000000.1	96.6	
*A. pseudocaelatus*	CBS 117616	39.7	STFS00000000.1	92.6	
*A. pseudonomius*	CBS 119388	37.8	STFR00000000.1	91.3	
*A. pseudotamarii*	CBS 117625	38.2	STFH00000000.1	92.5	
*A. sergii*	CBS 130017	38.3	STFL00000000.1	96.3	
*A. sojae*	NBRC 4239	39.8	BACA00000000.2	96.5	
*A. tamarii*	CBS 117626	38.5	STFJ00000000.1	92.1	
*A. transmontanensis*	CBS 130015	39.3	STFK00000000.1	96.5	
*A. oryzae* ^b^	ATCC 12892	41.2	NVQI00000000.1	96.5	

^a^: Percentages of DNA sequence identity compared to *A. flavus* NRRL3357 KA gene cluster sequence, respectively. Not significant, no/low significant similarity found. ^b^: The species was misidentified; see Results 3.1 and Discussion.

**Table 2 jof-09-00259-t002:** Partial KA gene clusters in *Aspergillus* and *Penicillium* species.

Species	KojR(%) ^a^	#AA	Protein ID ^b^	KojT(%) ^a^	#AA	Protein ID ^b^
*A. niger* CBS 513.88	100.0	561	XP_001393818.1 ^c^	100.0	572	XP_001393819.2 ^c^
*A. welwitschiae* CBS 139.54b	99.5	561	XP_026625778.1	99.0	585	XP_026625777.1
*A. phoenicis* ATCC 13157	99.1	561	RDK41302.1	99.3	585	RDK41301.1
*A. brasiliensis* CBS 101740	90.2	561	OJJ69434.1	94.1	572	OJJ69433.1
*A. tubingensis* CBS 134.48	88.8	560	OJI88168.1	91.5	585	OJI88169.1
*A. neoniger* CBS 115656	87.9	560	XP_025482490.1	93.2	567	XP_025482489.1
*A. luchuensis* CBS 106.47	87.7	560	OJZ82270.1	92.8	563	OJZ82269.1
*A. piperis* CBS 112811	87.5	560	XP_025520632.1	92.7	563	XP_025520631.1
*A. vadensis* CBS 113365	87.3	560	XP_025563047.1	93.2	567	XP_025563048.1
*A. eucalypticola* CBS 122712	87.0	560	XP_025388882.1	92.3	563	XP_025388881.1
*A. costaricaensis* CBS 115574	86.8	560	XP_025545016.1	93.4	572	XP_025545017.1
*A. sclerotioniger* CBS 115572	76.7	560	XP_025467076.1	85.7	574	XP_025467075.1
*A. carbonarius* ITEM 5010	75.9	559	OOF94303.1	85.7	659	OOF94304.1
*A. sclerotiicarbonarius* CBS 121057	74.9	558	PYI07967.1	85.3	571	PYI07968.1
*A. ibericus* CBS 121593	74.5	559	XP_025576212.1	86.9	573	XP_025576213.1
*A. transmontanensis* CBS 130015	62.9	555	KAE8316542.1	77.2	564	KAE8316543.1
*A. avenaceus* IBT 18842	61.7	553	KAE8154428.1	75.0	563	KAE8154429.1
*A. tanneri* NIH1004	55.8	545	XP_033424679.1	73.3	541	XP_033424678.1
*A. melleus* CBS 546.65	51.2	465	XP_045944549.1	73.2	545	XP_045944548.1
*A. steynii* IBT 23096	49.4	546	XP_024706771.1	73.7	546	XP_024706772.1
*P. nordicum* DAOMC 185683	100.0	544	KOS46106.1	100.0	548	KOS46105.1
*P. freii* DAOM 242723	90.8 ^c^	488	KUM59906.1	93.1 ^c^	548	KUM59902.1
*P. polonicum* IBT 4502	94.2 ^c^	292	OQD63438.1	92.5 ^c^	557	OQD63358.1

^a^: Percentages of amino acid sequence identity to the annotated KojR and KojT of *A. niger* CBS 513.88, respectively. ^b^: Consecutive numbers indicate that the genes encoding the proteins are situated next to each other. ^c^: Percentages of amino acid sequence identity to the annotated KojR and KojT of *P. nordicum* DAOMC 185683, which were treated as 100.0.

**Table 3 jof-09-00259-t003:** KA production of *kojR-* overexpressing transformants driven by the *gpdA* or *gpiA* promoter.

Strain	mg KA/g Dry Mycelia ^b^	Relative Amount
Control/(KuPG) ^a^	306.1 ± 187.0	1.00
D-8	304.0 ± 24.1	0.99
D-16	438.0 ± 57.8	1.43
D-20	337.8 ± 15.5	1.10
I-5	263.1 ± 35.2	0.86
I-9	342.0 ± 19.1	1.12
I-16	275.8 ± 39.2	0.90

^a^: KuPG is a *pyrG*+ parental strain of SRRC1709, which was the transformation recipient strain (see Materials and Methods). Cultures were grown at 30 °C for four days, and KA amounts in the supernatants were determined. ^b^: Numbers are averages of triplicate samples.

**Table 4 jof-09-00259-t004:** Comparison of zinc-finger DNA-binding domains of KojR and promoter regions of *kojA* and *kojT* of twenty-three aspergilli.

Species	Zinc-Finger Domain and Basic Region	KojR ^a^	*kojA* ^b^	*kojT* ^b^
*A. aflatoxiformans*	RAKRACETCKLRKRKCDGHEPCTYCLRYEYQCTFKPHPRRKPAASKS	100.0	100.0	99.2
*A. alliaceus*	RAKRACETCKLRKRKCDGHEPCTYCLRYEYQCTFKPHPRRRPAVPKN	91.7	100.0	67.4
*A. arachidicola*	RAKRACETCKLRKRKCDGHEPCTYCLRYEYQCTFKPHPRRKPAASKS	100.0	94.9	90.1
*A. bertholletius*	RAKRACETCKLRKRKCDGHEPCTYCLRYEYQCTFKPHPRRKPATSKS	96.7	82.5	77.7
*A. burnettii*	RAKRACETCKLRKRKCDGHEPCTYCLRYEYQCTFKPHPRRRPAVPKN	91.7	82.9	68.0
*A. caelatus*	RAKRACETCKLRKRKCDGHEPCTYCLRYEYQCTFKPHPRRKPAASKS	99.2	91.9	86.3
*A. coremiiformis*	RAKRACETCKLRKRKCDGHEPCTYCLRYEYQCTFNPHPRRKPAPTKS	91.7	63.9	60.5
*A. flavus_L*	RAKRACETCKLRKRKCDGHEPCTYCLRYEYQCTFKPHPRRKPAASKS	100.0	100.0	100.0
*A. flavus_S*	RAKRACETCKLRKRKCDGHEPCTYCLRYEYQCTFKPHPRRKPAASKS	100.0	100.0	100.0
*A. hancockii*	RAKRACETCKLRKRKCDGHEPCTYCLRYEYQCTFKPHPRRKPATSRS	95.8	77.3	65.1
*A. leporis*	RAKRACETCKLRKRKCDGHEPCTYCLRYEYQCTFKPHPRRKPAGSKS	95.8	75.1	66.3
*A. luteovirescens*	RAKRACETCKLRKRKCDGHEPCTYCLRYEYQCTFKPHPRRKPAASRS	98.3	87.4	85.6
*A. minisclerotigenes*	RAKRACETCKLRKRKCDGHEPCTYCLRYEYQCTFKPHPRRKPAASKS	100.0	98.9	97.4
*A. nomiae*	RAKRACETCKLRKRKCDGHEPCTYCLRYEYQCTFKPHPRRKPAASRS	95.8	88.4	83.6
*A. novoparasiticus*	RAKRACETCKLRKRKCDGHEPCTYCLRYEYQCTFKPHPRRKPTASKS	99.2	94.4	91.9
*A. oryzae*	RAKRACETCKLRKRKCDGHEPCTYCLRYEYQCTFKPHPRRKPAASKS	100.0	100.0	99.5
*A. parasiticus*	RAKRACETCKLRKRKCDGHEPCTYCLRYEYQCTFKPHPRRKPAASKS	100.0	95.4	91.9
*A. pseudocaelatus*	RAKRACETCKLRKRKCDGHEPCTYCLRYEYQCTFKPHPRRKPAASKS	99.2	92.3	86.3
*A. pseudonomius*	RAKRACETCKLRKRKCDGHEPCTYCLRYEYQCTFKPHPRRKPAASRS	95.8	88.6	84.7
*A. pseudotamarii*	RAKRACETCKLRKRKCDGHEPCTYCLRYEYQCTFKPHPRRKPAASKS	99.3	90.2	87.8
*A. sergii*	RAKRACETCKLRKRKCDGHEPCTYCLRYEYQCTFKPHPRRKPAASKS	100.0	94.3	92.7
*A. sojae*	RAKRACETCKLRKRKCDGHEPCTYCLRYEYQCTFKPHPRRKPAASKS	100.0	95.2	91.7
*A. tamarii*	RAKRACETCKLRKRKCDGHEPCTYCLRYEYQCTFKPHPRRKPTASKS	99.2	89.3	87.8
*A. transmontanensis*	RAKRACETCKLRKRKCDGHEPCTYCLRYEYQCTFKPHPRRKPAASKS	100.0	95.7	91.9

^a^: The sequence of the first 120 amino acids of the N-terminus was compared to that of *A. flavus*. Percentages of amino acid sequence identity are shown. ^b^: See Appendix A for promoter sequences of *kojA* and *kojT*, which were used for the MEME motif analysis. Percentages of nucleotide sequence identity are shown.

**Table 5 jof-09-00259-t005:** Locations of putative KojR-binding sites in the *kojA* and *kojT* promoters of the twenty-three aspergilli.

Species	* kojA * Promoter		* kojT * Promoter	
	Start/Strand	Motif Sequence	Start/Strand	Motif Sequence
*A. aflatoxiformans*	-277/+	CGACTTTGCCG	-205/+	CGGCTAAGTCG
*A. alliaceus*	-275/+	CGACTTTGCCG	-205/+	CGGCTATGTCG
*A. arachidicola*	-277/+	CGACTTTGCCG	-204/+	CGGCTAAGTCG
*A. bertholletius*	-278/+	CGACTTTGCCG	-204/+	CGGCTAAGTCG
*A. burnettii*	-276/+	CGACTTTGCCG	-205/+	CGGCTATGTCG
*A. caelatus*	-277/+	CGACTTTGCCG	-206/+	CGGCTAAGTCG
*A. coremiiformis*	-282/+	CGACTTTGCCG	-212/+	CGGGTAAGTCG
*A. flavus*	-277/+	CGACTTTGCCG	-205/+	CGGCTAAGTCG
*A. hancockii*	-298/+	CGACTTTGCCG	-202/+	CGGTTAAGTCG
*A. leporis*	-289/+	CGACTTTGCCG	-207/+	CGGCTAAGTCG
*A. luteovirescens*	-276/+	CGACTTTGCCG	-204/+	CGGTTAAGTCG
*A. minisclerotigenes*	-278/+	CGACTTTGCCG	-205/+	CGGCTAAGTCG
*A. nomiae*	-276/+	CGACTTTGCCG	-204/+	CGGCTAAGTCG
*A. novoparasiticus*	-278/+	CGACTTTGCCG	-204/+	CGGCTAAGTCG
*A. oryzae*	-277/+	CGACTTTGCCG	-205/+	CGGCTAAGTCG
*A. parasiticus*	-277/+	CGACTTTGCCG	-205/+	CGGCTAAGTCG
*A. pseudocaelatus*	-278/+	CGACTTTGCCG	-206/+	CGGCTAAGTCG
*A. pseudonomius*	-277/+	CGACTTTGCCG	-207/+	CGGCTAAGTCG
*A. pseudotamarii*	-278/+	CGACTTTGCCG	-175/+	CGGCTAAGTCG
*A. sergii*	-277/+	CGACTTTGCCG	-205/+	CGGCTAAGTCG
*A. sojae*	-277/+	CGACTTTGCCG	-204/+	CGGCTAAGTCG
*A. tamarii*	-278/+	CGACTTTGCCG	-204/+	CGGCTAAGTCG
*A. transmontanensis*	-277/+	CGACTTTGCCG	-204/+	CGGCTAAGTCG

These sites shown represent the 11-nucleotide imperfect palindromic sequence motifs found in the 5′ non-translated regions of respective genes. Negative number indicates the starting nucleotide position of each motif upstream of the ATG start codon (“A” treated as position zero). Strand designation “+” indicates the positive strand.

## Data Availability

Raw data and materials described in this study will be shared upon reasonable request, in accordance with USDA policies and procedures.

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
