# Peer review of "Kojic Acid Gene Clusters and the Transcriptional Activation Mechanism of Aspergillus flavus KojR on Expression of Clustered Genes"

_jof, 2023, doi:10.3390/jof9020259_

Round 1

Reviewer 1 Report

The manuscript by Chang et al. describes the distribution of the kojic acid gene cluster in section Flavi, other aspergilli, and a part of Penicillium species and the putative DNA-binding sequence of the transcription factor KojR present in the kojic acid gene cluster in Aspergillus flavus. It contains some scientifically interesting results, which would be worth being published. However, there are several flaws that should be appropriately revised.

Major points

1. The most serious flaw is the conclusion that kojA and kojT are regulated differently by KojR. The authors have drawn this conclusion from the results of the kojic acid production of the promoter mutants created by CRISPR/Cas9-mediated gene-targeting, but this conclusion is completely incorrect. As reported by Terabayashi et al. (Ref. 16), even if kojT was deleted, kojic acid production was not completely abolished, because kojA is intact and another unidentified transporter may function as a kojic acid efflux pump. Therefore, the authors cannot conclude that KojR is not involved in kojT gene expression just because the kojT promoter mutants produced kojic acid. The authors must examine the expression levels of kojT in the kojT promoter mutants.

2. The authors named the strains in which kojR driven by the gpdA or gpiA promoter was introduced into a kojR-disrupted strain as complemented strains, but this description is incorrect. A complemented strain is one in which the wild-type kojR gene is introduced, and the strains constructed by the authors are properly called overexpression strains. Also, why did they use two different promoters to overexpress kojR?

3. Fig. 2: Overexpression of kojR did not affect the amount of koji acid produced, which is quite strange. According to the report of Marui et al. (Ref.17), overexpression of kojR by using the tef1 promoter resulted in a significant increase in kojic acid production (Fig. 1A in Ref. 17). Thus, the result of this manuscript is not at all coincident with that of the Marui’s paper.

Minor points

1. Table 2 and line 374: According to the paper of Hong et al. (PLoS One, 8, e63769 (2013), Appl Microbiol Biotechnol, 98, 555-561 (2014)), the name A. awamori was misapplied and regarded A. awamori as doubtable species, and thus should be never used in classification.

2. Table 2: No explanation is provided for the superscript “c”.

3. Table 3: Explanation for the superscripts “a” and “b” is not required and should be deleted.

4. Table 4: Given that whole amino acid sequences of KojA and KojT were used for homology calculation, whole amino acid sequences of KojR should also be used for this purpose instead of N-terminal 120 amino acids.

5. Fig. 4A and B: No explanation is provided for the photographs of the agar plates. What does the CCG in red mean?

Author Response

Reviewer 1

The manuscript by Chang et al. describes the distribution of the kojic acid gene cluster in section Flavi, other aspergilli, and a part of Penicillium species and the putative DNA-binding sequence of the transcription factor KojR present in the kojic acid gene cluster in Aspergillus flavus. It contains some scientifically interesting results, which would be worth being published. However, there are several flaws that should be appropriately revised.

Major points

  1. The most serious flaw is the conclusion that kojA and kojT are regulated differently by KojR. The authors have drawn this conclusion from the results of the kojic acid production of the promoter mutants created by CRISPR/Cas9-mediated gene-targeting, but this conclusion is completely incorrect. As reported by Terabayashi et al. (Ref. 16), even if kojT was deleted, kojic acid production was not completely abolished, because kojA is intact and another unidentified transporter may function as a kojic acid efflux pump. Therefore, the authors cannot conclude that KojR is not involved in kojT gene expression just because the kojT promoter mutants produced kojic acid. The authors must examine the expression levels of kojT in the kojT promoter mutants.

Thank the reviewer for pointing out this. The result about kojT by Terabayashi et al. is quite intriguing. Since KojT transporter in theory does not possess enzymatic activity and thus could not directly involved in KA production. Defect/loss of KojT would not affect amounts of KA produced by KojA oxidoreductase or other to-be-identified KA enzymes. Another reviewer (#3) also raises some doubt about the role of KojT in KA biosynthesis. In other words, its defect should not affect KA production. However, as commented, the following text has been included in the discussion section. The statement “kojA and kojT are regulated differently by KojR.” in Abstract and Discussion has been deleted. The manuscript is now focused on the significance of the KojR-binding site in the kojA promoter as described in Abstract. The inadequacy of the present work also is mentioned at the end of the Discussion section. I hope these changes are acceptable to the reviewer.

“However, in the characterization of A. oryzae kojT by Terabayashi et al. [16], the kojT deletion mutant still produced a reduced amount of KA as evidence by the colony showing a pale red color. Therefore, we cannot rule out that KA production in these A. flavus mutants results from the presence of another unidentified transporter protein that can replace the function of KojT as previously proposed.”

  1. The authors named the strains in which kojR driven by the gpdA or gpiA promoter was introduced into a kojR-disrupted strain as complemented strains, but this description is incorrect. A complemented strain is one in which the wild-type kojR gene is introduced, and the strains constructed by the authors are properly called overexpression strains. Also, why did they use two different promoters to overexpress kojR?

As commented, the term “complemented” has been revise to “overexpression” throughout the text. In the work by Marui et al, the A. oryzae tef1 (translation elongation factor EF-1 alpha) gene promoter was used to drive kojR expression. To further that aspect, both homogenous and heterologous promoters were used to examine whether the native A. flavus kojR promoter is dispensable in term of possible auto-regulation. Furthermore, using both promoters was to demonstrate that transcription activation indeed is involved in KA biosynthesis even when kojR expression is driven by different promoters.

  1. Fig. 2: Overexpression of kojR did not affect the amount of koji acid produced, which is quite strange. According to the report of Marui et al. (Ref.17), overexpression of kojR by using the tef1 promoter resulted in a significant increase in kojic acid production (Fig. 1A in Ref. 17). Thus, the result of this manuscript is not at all coincident with that of the Marui’s paper.

As to the study published by Marui et al., their main purpose was to show that KA biosynthesis is regulated by a transcriptional activation mechanism as the present study. Their results about KA production was based on culture supernatants (not normalized to mycelial dry weight). In other words, they assumed that all cultures produced same amounts of mycelia, which likely is not true. Also, only one kojR-overexpressing strain was (selected) used and no information about its genetic background (i.e., copy number) was given. It is highly likely that if other putative kojR-overexpressing strains were used, they would have given different results in term of KA production, especially when introduced vectors integrated into different locations in the genome. Although their experimental design was not ideal, it is sufficient for readers to understand the main idea behind the work. In other words, absolute numbers for the amounts of KA produced are not that critical because as we know even slight changes in culturing conditions can affect the amounts of KA produced. In the present work, three independent putative kojR-overexpression strains from each promoter set were selected and analyzed to give a better representation. In the originally presented Table 3, an outlier in the control (triplicate determinations) was not included (a higher average for the control), which resulted in lower relative ratios. Nonetheless, closely examining the previous data, readers would find D-16 produced more KA than the control. To respond to the reviewer’s comment, three determinations from the control were included in the revised Table 3, and it now shows that D-16, D-20, and I-9 produced more KA than the control although the increased levels by the latter two were to a lesser degree.

Minor points

  1. Table 2 and line 374: According to the paper of Hong et al. (PLoS One, 8, e63769 (2013), Appl Microbiol Biotechnol, 98, 555-561 (2014)), the name A. awamori was misapplied and regarded A. awamori as doubtable species, and thus should be never used in classification.

Thank you for pointing out this information. The molecular classification of black molds by Hong et al. was solely based on β-tubulin and calmodulin gene sequences (PLoS One) and latter discussed in a mini-review (AMB). The multi-locus sequence typing technique had been conventionally and routinely employed before the advent of the genomic era. A. awamori along with other apparently belong to Aspergillus section Nigri. However, I don’t have the expertise in judging the classification of black molds. The use of Aspergillus species names is strictly based on those available from and published by NCBI. Therefore, deciphering the phylogenomic relationship between A. awamori and others better be left to other black-mold research groups that can reach a consensus about the correct identity/classification of A. awamori through their future studies.

  1. Table 2: No explanation is provided for the superscript “c”.

Note for “c” was accidentally omitted when transferring Table 2 to the final version. As commented, it has been included in the revised manuscript as below.

c: Percentages of amino acid sequence identity to annotated KojR and KojT of P. nordicum DAOMC 185683, which were treated as 100.0, respectively.

  1. Table 3: Explanation for the superscripts “a” and “b” is not required and should be deleted.

As commented, both have been deleted in the revised manuscript.

  1. Table 4: Given that whole amino acid sequences of KojA and KojT were used for homology calculation, whole amino acid sequences of KojR should also be used for this purpose instead of N-terminal 120 amino acids.

Table 4 summarizes the degree of DNA homology of putative promoter regions (not gene sequences) of kojA and kojT as supposedly shown in Table S2. But I made a mistake and forgot to add supplemental Tables S1 and S2 to the Supplemental materials (only four supplemental figures were submitted). I have added both Tables S1 and S2 to the revised Supplemental materials. The reason for only showing the N-terminal 120 amino acids is that this portion of a C6-type regulatory protein usually contains its Zn(II)2Cys6 DNA-binding domain (binding to regulated gene promoters) and a basic amino acid region. A new footnote under Table 4 has been added as below.

b: See Table S2 for promoter sequences of kojA and kojT, which were used for the MEME motif analysis. Percentages of nucleotide sequence identity are shown.

  1. Fig. 4A and B: No explanation is provided for the photographs of the agar plates. What does the CCG in red mean?

The design of a CRISPR/Cas9 gene target site always uses a triplet called protospacer adjacent motif, which is NGG. This motif can be on either (positive or negative) strand. In the case of the kojT gene knockout, it is on the positive and is ---CGG. In the case of kojA gene knockout, it is on the negative strand and shown as CCG--- (on the positive strand, it is also ---CGG).

As commented, the following explanations have been added to the figure legend.

“The photos above the sequences are colony morphologies of six mutants on KAM agar plates, which are shown alternatively front and reverse sides. Colonies were grown at 30 ° for five days in the dark. Orange red plates are KA-producing colonies.”

“Red trinucleotides CCG and CGG are protospacer adjacent motifs (PAM) that follow the regions targeted for cleavage by the Cas9 nuclease.”

Reviewer 2 Report

KA has important applications in the cosmetics and food industries. Based on the analysis of multiple KA_BGCs, the authors identified a segment of the palindrome sequence present in the HojR region. This sequence was further developed using CriSPS/cas9 technology to develop a targeted regulation technique to achieve precise regulation of KA_BGC. Overall, it is a good work.

As the authors described at the beginning of the abstract that KA has important applications in the cosmetic and food industries, then this work is directly related to KA biosynthesis, and it is suggested to add a relevant description at the end of the abstract to enhance the significance of this manuscript.

Line 286. What does NJ indicate ?

Figure 2, The mark ABC font size is not consistent.

Figure 4A&B, The morphological photos are not clear enough, and also what is the purpose of these morphological pictures here?

Author Response

Reviewer 2

Comments and Suggestions for Authors

KA has important applications in the cosmetics and food industries. Based on the analysis of multiple KA_BGCs, the authors identified a segment of the palindrome sequence present in the HojR region. This sequence was further developed using CriSPS/cas9 technology to develop a targeted regulation technique to achieve precise regulation of KA_BGC. Overall, it is a good work.

As the authors described at the beginning of the abstract that KA has important applications in the cosmetic and food industries, then this work is directly related to KA biosynthesis, and it is suggested to add a relevant description at the end of the abstract to enhance the significance of this manuscript.

As comment, the statement “The information gained may facilitate strain improvement and benefit future kojic acid production.” has been added to Abstract.

Line 286. What does NJ indicate ?

NJ (Neighbor-Joining) is a bottom-up (agglomerative) clustering method commonly used in bioinformatics for the creation of phylogenetic trees.

Figure 2, The mark ABC font size is not consistent.

The pasted photo images are scalable. It can be adjusted accordingly. I have scaled up “B” in the revised manuscript to make the appearance consistent with others. I also have replaced “A” because of the font color of the previous version was not consistent.

Figure 4A&B, The morphological photos are not clear enough, and also what is the purpose of these morphological pictures here?

The purpose of these photos is to show whether the gene knockout mutants produce kojic acid or not. The orange red color is the typic color resulting from the kojic acid-iron complex. As commented, the following explanation has been added to the revised manuscript.

The photos above the sequences are colony morphologies of six mutants on KAM agar plates, which are shown alternatively front and reverse sides. Colonies were grown at 30 ° for five days in the dark. Orange red plates are KA-producing colonies.

Reviewer 3 Report

In the manuscript entitled "Kojic acid gene clusters and the transcriptional activation mechanism of Aspergillus flavus KojR on expression of clustered genes" the authors present the results of their studies on the presence and conservation of the tripartite Kojic acid (KA) biosynthetic gene cluster in Aspergillus section Flavi species. Generally, the introduction reads well. However, a scheme illustrating the biosynthesis of KA and the steps involving the cluster proteins would be informative. The authors aimed to identify and confirm the binding sites of the KA cluster-specific regulator KojR in the promoters of the the two other cluster genes via multiple sequence alignments and CRISPR-Cas9-based mutagenesis, respectively. In addition deletion and complementation of the kojR gene was performed. However, in my opinion the experimental design is not ideal for the use of a ku70 mutant (SRRC1709 is ku70∆ according to PMID 31416879). 

Major points:

For the complementation studies, how can the authors infer the length of a promoter from RNAseq read coverage?

I wonder how the complementation cassettes should integrate into the genome of the ku70∆ recipient strain that lacks the non-homologous end joining pathway (NHEJ). Have the authors checked if the Afl-gpiA promoter constructs integrate at the Afl-gpiA locus? Where would the heterologous gpdA constructs integrate? Are the multiple copy numbers caused by tandem integrations? These questions might be answered with Southern blotting experiments. 

Generation of deletion and complementation cassettes cannot be reconstructed, because supplementary Tables are missing in the submission. Which part of kojR was deleted?

Figure 2: 

Panel B: Why is the expression of kojR in the complementation strains higher after 48 h than after 72 h? Shouldn't the used promoters drive expression constitutively? Or does gpiA need special induction and therefore a different growth medium? Why has so much iron to be included in the medium? If expression of kojR in the wild type would be higher after 72 h compared to 48 h, this should be shown. The expression after 48 and 72 h should be displayed separately.

Panel C: Expression of kojA and kojT does not seem to correlate at all to kojR expression (compare D-8 vs I-5 and I-9), is there a way to explain this? How can the huge variability of gene expression be explained? Why is expression normalized to ∆kojR and not to the control as in Figure 2B?

Figure 2D is mentioned in the text but not marked/shown in the figure.

I don't agree that "the KA amounts produced by each set reflected the overall gene expression levels of kojA and kojT at 72 h" (Lines 347ff). I-9 shows the lowest expression but still has >80% of the control KA production. Why is KA quantified after 96 h and not after 72 h? The growth time should be included in the caption of Table 3. 

Figure 3 does not correspond to Table 5. It seems like the motifs of the kojA and the kojT promoters were pooled, which results in a somehow misleading sequence logo. When having a closer look at the provided motif sequences, no reversed repeats are detectable at the ends of the motifs – CGA....CCG and CGG....TCG in kojA and kojT, respectively. If just the sequences of Table 5 were used for the logo, shouldn't there be a single T at position 5? Which sequences have actually been included for the generation of the logo? Furthermore, what happened to the sequence logo display? Shouldn't single letters (pos 1,2,10,11) reach the bottom line of the logo? The information of Table 5 is redundant regarding the displayed species, as is the case for Table 4. Therefore, these tables should be moved to the supplemental.

CRISPR/Cas9 and Figure 4:

What do the authors mean by "location specific primers"? Since supplemental tables are missing, I was not able to check the annealing sites. Hoever, a scheme would be nice for the reader. How can the lack of a positive PCR result be explained in the light of a KA minus phenotype? The colonies displayed at the top of the sequence alignments are what? Do they refer to the mutants? Are top and bottom views shown? How old are the displayed colonies? This should be indicated in the figure caption. The wavy red lines should be removed.

Since a strain lacking NHEJ was used and no repair template was provided for micro-homology-based repair, it seems strange that so many strains containing indels could be identified.

Generally, the figure and table captions have to be revised to contain all the information needed to be able to understand figures and tables on their own.   

I don't understand how the first paragraph of the discussion section relates to the present study. ATCC12892? This should be clarified or removed.

I don't agree that "the CRISPR/Cas9 technology indeed could serve as an alternative to the ChIP-seq method…". I rather would regard this strategy to be complementary but not an alternative to the mentioned techniques. 

In my opinion, the main question is how the loss of a transporter should result in a lack of metabolite production? Therefore, the presence of KA production in the kojT motif mutants does not necessarily exclude the possibility of a loss of KojR binding and abolished kojT expression. 

Instead of just referring to KA production, gene expression of kojA, kojR and kojT should be analyzed in the CRISPR mutants.

Minor points: 

Table 1 should be moved to the results section or to the supplementary. What does "not significant" mean regarding sequence identity?

Line 144: AspGD is offline. Please refer to and cite fungiDB and/or ENSEMBLfungi instead.

Line 154f: I could not find AFLA_113120 in any database, instead AFLA_007258 should be the accession for gpiA in current releases. 

PMID 322111196 should be cited here as well. It describes the use of the gpiA promoter for expression of genes.

Lines 228ff: It seems that plasmids 191015 and 191016 are no longer available from Addgene?

Line 270: The current accession numbers of kojA, kojR and kojT should be included.

Figure 1: What does "NJ analysis" refer to? Please indicate what is shown by the scale bar. It seems that ATCC12892 is not included in Table 1, why not? It would be good to include the strain names after the species name to exclude any ambiguities.

Table 2: In the footer, footnote "b" might refer to "c" in the table and description of footnote "b" seems to be missing? Please check. 

Line 328: By reading the manuscript I was unsure about the correct use of "respectively" several times. Here, I am pretty sure that "respectively" should be removed.

Table 3: "Control/KUPG" should probably be replaced with "Control (KuPG)". It should be indicated in the footer, what the numbers below "kojA" and "kojT" refer to. Sequence identity? For which region? There also is a "basis region" in one of the column headers.

Author Response

Reviewer 3

In the manuscript entitled "Kojic acid gene clusters and the transcriptional activation mechanism of Aspergillus flavus KojR on expression of clustered genes" the authors present the results of their studies on the presence and conservation of the tripartite Kojic acid (KA) biosynthetic gene cluster in Aspergillus section Flavi species. Generally, the introduction reads well. However, a scheme illustrating the biosynthesis of KA and the steps involving the cluster proteins would be informative.

Thank the reviewer for this comment, it is a good suggestion, but the request cannot be fulfilled based on current knowledge. As stated in the Introduction section “….The precise nature of the KA biosynthetic process remains virtually unclear”. In other words, despite that KA biosynthesis was first studies in early 1950s by Dr. Bentley’s group, and pathways proposed in 1981 by Bajpai et al., not much experimental evidence has been obtained since then. Kojic acid studies in the past 70 years mostly had been focused on various aspects of production. Although the three-gene kojic acid biosynthesis gene cluster was identified in 2010, it likely is an incomplete gene cluster; more work is needed to identify other unknown genes involved in other to-be-resolved conversion steps. Of the three genes (kojA-kojR-kojT), kojA was annotated as encoding an oxidoreductase and, in theory, should be directly involved in kojic acid formation, but the precursor it catalyzes is not known. The other two genes, kojR and kojT, which encode a Zn(II)2Cys6 zinc cluster regulator and a transporter protein (efflux pump), respectively. These two gene products apparently do not possess enzymatic activities and hence are not involved in any metabolite conversion steps.

The authors aimed to identify and confirm the binding sites of the KA cluster-specific regulator KojR in the promoters of the the two other cluster genes via multiple sequence alignments and CRISPR-Cas9-based mutagenesis, respectively. In addition deletion and complementation of the kojR gene was performed. However, in my opinion the experimental design is not ideal for the use of a ku70 mutant (SRRC1709 is ku70∆ according to PMID 31416879).

The reviewer is correct that SRRC1709 is a nonhomologous end-joining (NHEJ)-deficient strain, of which the ku70 gene is disabled. The reviewer’s comment about the unsuitability of this strain probably comes from the perception that fungi and other organisms only have two repair systems, i.e., NHEJ and homologous recombination (HR). This perception still have room for debate. As a matter of fact, since the creation of SRRC1709 (derived from wild-type CA14) in 2010, roughly 90% of A. flavus gene knockout and subsequent gene complementation experiments have used this strain (another available but less frequently used ∆ku70 strain is derived from NRRL3357). The use of an NHEJ strain facilitates the generation of correct gene knockout mutants via the so-called double-crossover recombination mechanism (the transforming DNA often is a piece of linear DNA that contains two homologous flanking sequences to the targeting gene with a selection marker in between). This method is quite similar to what the reviewer later mentioned “micro-homology-based repair”, which also relies on homologous flanking sequences to the CRISPR/Cas9 target site. However, the obtained frequencies of gene disruption with a NHEJ-deficient recipient strain are not always 100%, and most commonly range from 60% to 80%. In other words, even under an NHEJ genetic background (supposedly only HR is operating) and supposedly only one location in the genome can the transforming DNA integrate, a portion of the primary transformants are not knockouts. This result implies that the transforming DNA (selection marker and/or flanking sequences) integrates into heterologous sites and that unknown repair mechanism(s) in addition to HR or other than NHEJ and HR somehow is involved. Therefore, even though we haven’t fully understood the science behind it, the use of SRRC1709 should not be an issue as commented.  Furthermore, a recent study has shown that NHEJ-deficient mutants appear to be genetically stable. Please see the article “Genome sequencing of evolved aspergilli populations reveals robust genomes, transversions in A. flavus, and sexual aberrancy in non-homologous end-joining mutants. BMC Biology (2019) 17:88”.

Major points:

For the complementation studies, how can the authors infer the length of a promoter from RNAseq read coverage?

If the reviewer is familiar with RNAseq work, the reviewer might have used a genome browser like JBrowse (https://jbrowse.org/jb2/) to align reads to a reference genome sequence, of which genes have been annotated (start, end, introns etc.). Since in between annotated genes, a blank region (no reads, i.e., no mRNAs) exists. By looking at the coverage and the spread of reads beyond/upstream of the ATG start codon, the putative promoter region can be estimated. I need to emphasize that this putative region is just an educational guess, no one in the research world really knows for sure what the exact promoter region for a gene is. It is defined by current experimental evidence and will be revised when new knowledge becomes available.

I wonder how the complementation cassettes should integrate into the genome of the ku70∆ recipient strain that lacks the non-homologous end joining pathway (NHEJ). Have the authors checked if the Afl-gpiA promoter constructs integrate at the Afl-gpiA locus? Where would the heterologous gpdA constructs integrate? Are the multiple copy numbers caused by tandem integrations? These questions might be answered with Southern blotting experiments.

As explained by my reply to the second comment regarding the perception about the NHEJ-deficient (∆ku70) recipient strain, and I hope some doubt has been cleared from the reviewer’s mind. The question about how exactly a transforming DNA/vector integrates into a genome is really an intriguing one. It is still an unresolved issue despite that great numbers of gene functions studies have been carried out. In terms of integration sites, to accurately determine them based on currently available molecular techniques will be a daunting task and likely will not generate any meaningful conclusions. Although using some long-read genome sequencing approaches might provide valuable information, but it’s time-consuming and is not cost effective. A case in point, of over 100 gene function studies of A. flavus (often included complementation strains) that have been carried, I don’t remember I have ever seen researchers investigating how exactly introduced genes integrated into the genomes of the knockout strains. It’s just too complicated to find an answer. Restored wild-type phenotypes or gene expression are often used as criteria for genetic complementation. Conceptionally, for a circular DNA (vector) its integration mechanism may differ from that of a linear DNA, for example, the gene disruption cassette (with two homologous flanking sequences) mentioned earlier. In addition to homologous regions such as the gpiA promoter and the remaining kojR portion in the kojR knockout, integration of gpiA- and gpdA-containing vectors into other genome regions are possible. The reviewer mentioned “micro-homology-based repair”, and it seems that the “micro” may be equivalent to 20-40 nucleotides in length. A. flavus genome is about 37 Mb (8 chromosomes). It is likely that micro/small homologous regions other than the gene and promoter sequences exist in the fungal genome. A recombinational event by the “single-crossover mechanism” would allow a circular vector to integrate into the genome. The reviewer can refer to an early PNAS article, Yeast transformation: a model system for the study of recombination, 1981, 78: 6354. I personally think that under the selection pressure of either harmful drug or nutrition depletion, for an organism to survive, the probability and need for this type of single-crossover integration should increase significantly.

Southern hybridization is an old-fashion technique, its success depends on restriction patterns around an integration site are known without ambiguity. Hence the selection of restriction enzymes as well as the region used for generating a probe affects the results greatly. Using this technique likely will not reach conclusive results.

Generation of deletion and complementation cassettes cannot be reconstructed, because supplementary Tables are missing in the submission. Which part of kojR was deleted?

I apologize for not including Table S1 that shows primer sequences and associated notes in previous submission. It has been added to the revised Supplemental materials. The deleted region is from about 0.2 kb upstream of the ATG start site plus 0.8 kb of the coding region (about half) which includes the coding sequence for the Zn(II)2Cys6 DNA-binding motif and the putative dimerization.

Figure 2:

Panel B: Why is the expression of kojR in the complementation strains higher after 48 h than after 72 h? Shouldn't the used promoters drive expression constitutively? Or does gpiA need special induction and therefore a different growth medium? Why has so much iron to be included in the medium? If expression of kojR in the wild type would be higher after 72 h compared to 48 h, this should be shown. The expression after 48 and 72 h should be displayed separately.

Changes in Panel B are results from the time-course comparison of kojR expression levels of the kojR-complemented (overexpression) strains to those of the control strain. The results are not normalized expression levels. The sole purpose of this panel B is to show that the promoters (gpdA and gpiA) indeed can drive higher expression of kojR at different time points when compared to the control. To clarify this point and possible other questions the reviewer has, I have prepared and added a new supplemental table (Table S3) to the Supplemental Materials that shows normalized gene expression levels including changes from 48 h to 72 h of all strains. To answer the reviewer’s question about kojR expression of the control at 48 h and 72 h- as the reviewer can see from Table S3, the expression level is higher (1.6-fold) at 72 h.

Thank the reviewer for pointing out the incorrect concentration. it was a typo and should be 1.0 mM as the original recipe of KAM not 10 mM.

Panel C: Expression of kojA and ojT does not seem to correlate at all to kojR expression (compare D-8 vs I-5 and I-9), is there a way to explain this? How can the huge variability of gene expression be explained? Why is expression normalized to ∆kojR and not to the control as in Figure 2B?

Unfortunately, this is the nature of time-course experiments (and transcriptomic work based on reads) although I have seen quite a few researchers presenting well correlated data, which mostly are from using one strain each. Gene expression is determined and affected by factors more than we can imagine. A case in point, even an experiment is repeated under supposedly the same conditions, the obtained data usually are different, sometimes with great variations. Also, for the same qRT-PCR experiment, if different normalizer is used (18S, β-tubulin or other house-keeping genes), the relative fold-changes often are different.

Special and temporal gene expression is intricate and complicate. Regulatory genes and structural genes are also controlled differently. A possible “lapse in time” scenario (based on the concept of secondary metabolism) is that the regulatory gene kojR is expressed first (18-24 h) and the level reaches a maximum at 48 h and fluctuates thereafter but seems to be tightly regulated (as seen at 72 h in Table S3). The KojR protein is later (at 48-72) produced and binds to transactivate expression of structural genes of kojA and kojT, their levels thus increase from the basal levels (likely before 48 h) and much higher expression levels are detected at 72 h. Subsequently, enzymes (mostly unknown yet) for kojic acid formation are synthesized to convert precursors and intermediates to the final product. Most kojic acid is produced and accumulated at 96 h, a reason to determine kojic acid at 96 h, simply to give readers an idea about the KA producing capacities of the complementation/overexpression strains.

The reason for using the kojA and kojT expression levels of the ∆kojR strain is that these levels in theory represent basal expression levels, i.e., the baselines (relative fold treated as 1) for comparison with those after induction (gene expression transactivated by introduced kojR gene/expressed KojR protein). In the control strain, which has an intact kojR gene (KojR regulator), the transactivational mechanism is operating, which means that levels detected at 48 h and 72 h are after induction not the supposed basal levels. In short, it’s more logic to compare kojA and kojT levels before (basal levels) and after (transactivated) induction in the same strain (∆kojR) than comparing with the control since vectors with different promoters (gpdA and gpiA) were introduced into the ∆kojR strain, not the control strain.

I have incorporated data presented in Table S3 in the revised Results section and rewritten a big portion of it. I hope this will make it easier for understanding how overexpressing kojR affects kojR expression and the effect of the kojR expression in increasing expression of kojA and kojT, which eventually leads to KA production/accumulation in the genetically complemented ∆kojR strains.

Figure 2D is mentioned in the text but not marked/shown in the figure.

Thank the reviewer for pointing out this. Figure 2D is part of Figure 2C. It has been corrected.

I don't agree that "the KA amounts produced by each set reflected the overall gene expression levels of kojA and kojT at 72 h" (Lines 347ff). I-9 shows the lowest expression but still has >80% of the control KA production. Why is KA quantified after 96 h and not after 72 h? The growth time should be included in the caption of Table 3.

The statement is merely a generalization of the result. Since it may cause confusion to readers, it has been deleted.

As suggested, the growth conditions in now included as a footnote in Table 3.

Figure 3 does not correspond to Table 5. It seems like the motifs of the kojA and the kojT promoters were pooled, which results in a somehow misleading sequence logo. When having a closer look at the provided motif sequences, no reversed repeats are detectable at the ends of the motifs – CGA....CCG and CGG....TCG in kojA and kojT, respectively. If just the sequences of Table 5 were used for the logo, shouldn't there be a single T at position 5? Which sequences have actually been included for the generation of the logo? Furthermore, what happened to the sequence logo display? Shouldn't single letters (pos 1,2,10,11) reach the bottom line of the logo? The information of Table 5 is redundant regarding the displayed species, as is the case for Table 4. Therefore, these tables should be moved to the supplemental.

I apologize for forgetting to include Table S2 in previous submission, which caused the confusion. It has been added to the revised Supplemental Materials. Table S2 contains sequences in the putative promoter regions of kojA and kojT for the MEME motif analysis (input). Table 5 is the summary of the derived putative KojR-binding sequences of kojA and kojT from the 23 aspergilli that have a complete KA gene cluster (results). By the same token, Table 4 is the summary, which is presented to give readers information about the zinc-finger DNA-binding domains of KojR and degrees of nucleotide identity of promoter regions of kojA and kojT of 23 aspergilli.

The logo is the derived “pattern” based on the newly included Table S2 sequences not based on Table 5. As to the displayed logo, it appears that the reviewer has a keen sight and is meticulous. I would say that 99% publications that contain MEME-motifs use the pattern “reach the bottom” described by the reviewer. However, the reviewer can find an exception in Figure 2 of the 2019 article by Ullah et al. Genome-wide identification and evolutionary analysis of TGA transcription factors in soybean. Sci Rep 9, 11186. The reason for not “reach the bottom” is as follows-

In MEME’s logo output (downloadable), there are two formats- one is the PNG (for web) version; this is the bit map that has the same baseline for all positions and seen on the desktop/laptop screen and apparently commonly presented by researchers. Another format is the EPS (for publication) version, which is the format presented in this work. Although their appearances are somewhat different, in essence, they are the same, i.e., the relative height is how certain to observe a particular nucleotide at a particular position is and high heights indicate high conservation/low uncertainty.

As commented, an explanation has been added to the figure legend to avoid confusion to some readers. “The logo is the downloaded EPS (for publication) version from the MEME site, which in appearance is somewhat different from the PNG (for web) version that all positions have the same baseline. The relative height is how certain to observe a particular nucleotide at a particular position is and high heights indicate high conservation/low uncertainty.”

CRISPR/Cas9 and Figure 4:

What do the authors mean by "location specific primers"? Since supplemental tables are missing, I was not able to check the annealing sites. Hoever, a scheme would be nice for the reader. How can the lack of a positive PCR result be explained in the light of a KA minus phenotype? The colonies displayed at the top of the sequence alignments are what? Do they refer to the mutants? Are top and bottom views shown? How old are the displayed colonies? This should be indicated in the figure caption. The wavy red lines should be removed.

I apologize for forgetting to include Table S1; it has been added to the revised Supplemental Materials. The reviewer now should be able to easily examine the annealing sites based on the primer sequences (and KA gene designations) provided. As a matter of fact, CRISPR/Cas9 systems have been developed and used for so many years, most publications now no longer mention details about the PCR-sequencing part because it’s somewhat self-evident. In general, the way it is done is to PCR with “location specific primers”, this usually means designing primers based on sequences approximately 0.5 kb upstream and downstream of a gene target site (distance/positions can vary). The 1.0 kb or so PCR product generated is the most suitable size for Sanger sequencing (one pass, normal read) to identify mutated regions. Although seldomly reported in early fungal CRISPR/Cas9 studies, it is now realized as reported by some recent publications that large deletions in addition to small indels can be created and are not uncommon. In A. flavus, it is quite common to have deletions larger than 1.0 kb in size. Those KA mutants that did not give PCR products presumptively contain large deletions beyond the primer sites.

As commented, revisions/correctios have been made to the figure legend accordioning.

“The recipient strain used for the CRISPR/Cas9 work is wild-type A. flavus CA14. Wt represents wild-type sequences.”

“Red trinucleotides CCG and CGG are protospacer adjacent motifs (PAM) that follow the regions targeted for cleavage by the Cas9 nuclease.”

“The photos above the sequences are colony morphologies of six mutants on KAM agar plates, which are shown alternatively front and reverse sides. Colonies were grown at 30 ° for five days in the dark. Orange red plates are KA-producing colonies.”

“…in the kojA and kojR intergenic region inferred from the present study. The site is 266 nucleotides from the translation start codon of kojA and 466 nucleotides from the start codon of kojR.”

Since a strain lacking NHEJ was used and no repair template was provided for micro-homology-based repair, it seems strange that so many strains containing indels could be identified.

The recipient strain used for the CRISPR/Cas9 system is wild-type A. flavus CA14 not the NHEJ-deficient strain used in gene knockout and complementation experiments. It was mentioned in M&M subsection 2.10 “Gene-targeting vectors were transformed into wild-type A. flavus CA14 recipient as previously described.” The reviewer is correct that using a NHEJ-deficient strain, primary transformants would barely be generated without donor DNA because most (or all) mutations are lethal. A reason for this likely is that the genome is constantly under the cutting pressure (or action) of Cas9 nuclease (the molecular scissors) and no donor DNA is available for repairing the damages.

For easy reference, a statement “The recipient strain used for the CRISPR/Cas9 system is wild-type A. flavus CA14” has been added to the figure legend.

Generally, the figure and table captions have to be revised to contain all the information needed to be able to understand figures and tables on their own.

Thank the reviewer for this comment. I have made every effort to improve the readability. Please see the revised version.

I don't understand how the first paragraph of the discussion section relates to the present study. ATCC12892? This should be clarified or removed.

The reviewer probably is working on other organisms not on fungi. Misclassification of Aspergillus species has been a concern especially after the advent of the genomic era. The reviewer may refer to a recent article by Arias, et al - Aspergillus section Flavi, Need for a robust taxonomy. Microbiol Resour Announc (2021) 10, e0078421. “Section” is a taxonomic rank between genus and species. Classification of Aspergillus species conventionally uses the multi-locus sequence typing approach. Because the work started with a general survey of KA gene clusters in fungi, it is just a natural extension of showing the utility of the KA gene cluster information, i.e., using these sequences to decipher phylogenetic relationships among section Flavi aspergilli and to correct misidentified Aspergillus isolates. An explanation has been added to the Discussion in the revised manuscript.

I don't agree that "the CRISPR/Cas9 technology indeed could serve as an alternative to the ChIP-seq method…". I rather would regard this strategy to be complementary but not an alternative to the mentioned techniques.

The reviewer is right about this. Since the CRISPR/Cas9 technology indeed cannot replace the ChIP-seq method and the conclusive evidence of KojR-binding to kojA or kojT promoters, i.e., the identification of genuine motif sequence(s) can only come from ChIP-seq results. The word “complementary” has been used.

In my opinion, the main question is how the loss of a transporter should result in a lack of metabolite production? Therefore, the presence of KA production in the kojT motif mutants does not necessarily exclude the possibility of a loss of KojR binding and abolished kojT expression.

The reviewer’s thought is logical as to “…KA production in the kojT motif mutants does not necessarily exclude the possibility of a loss of KojR binding and abolished kojT expression”. As to the kojT function, Terabayashi et al. (Identification and characterization of genes responsible for biosynthesis of kojic acid, an industrially important compound from Aspergillus oryzae. Fungal Genet Biol 2010) show that disruption of kojT in Aspergillus oryzae significantly affects KA production (mutant colony exhibiting pale red phenotype) although it does not abolish the production. The authors thus proposed that the presence of another unidentified transporter protein that can replace the function of KojT transporter. The interpretation of the current result was based on their results, i.e., kojT gene/KojT transporter somehow is involved in KA production (although it seems unlikely) assuming that if kojT expression in A. flavus was affected, the same phenotypic change (a significant reduction in KA-iron complex pigmentation) should be seen. However, no such change was observed from the nine different types of kojT deletion mutants, and they display the same extent of orange-red pigmentation as the wild-type strain. Another reviewer (Reviewer #1) also believes the presence of an unidentified transporter with the same function as KojT is the reason for the current results of KA production. If as the reviewer thinks that KojT transporter protein has no bearing on KA production because it is just an efflux pump and does not possess enzymatic activity (I do agree with this notion), the results would have to be interpreted as kojT is not specifically regulated by KojR. To take into consideration of both opinions, I have removed the statement “kojA and kojT are regulated differently” in Abstract and Discussion and focus only on the kojA part that can be satisfactorily explained.

Instead of just referring to KA production, gene expression of kojA, kojR and kojT should be analyzed in the CRISPR mutants.

This is a good suggestion. However, because the direct involvement of kojT in KA production is not clearly defined as described above and the lack of technical support for a detailed study, it will have to wait for other research groups that are interested in KA biosynthesis and regulation to include those currently unknown KA structural genes in their studies so that meaningful conclusions can be drawn. Two of the co-authors have retired. I will soon conclude my 35 years’ research career and no longer have technical support in the lab. I believe the information derived from the six types of kojA mutants that lost KA production is significant because since the incomplete KA gene cluster was discovered over a decade ago, not much progress has been made in terms of its regulation. However, I also pointed out the inadequacy of the current work at the end of the Discussion section. I hope these explanations are acceptable to the reviewer.

“Despite the initial characterization of KojR-binding sites in the present study, the final proof about the identity of the KojR-binding motif including its actual length will have to come from direct evidence of protein-DNA interaction, such as the use of ChIP technology. Future characterization of those yet to-be-identified structural genes that are directly involved in KA formation will shed light on genuine KojR-binding sequences in the KA biosynthesis gene promoters.”

Minor points:

Table 1 should be moved to the results section or to the supplementary. What does "not significant" mean regarding sequence identity?

Table 1 previously submitted was placed at the end before the reference section along with other Tables because of its width. Manuscript formatting was done by the journal staff. It is a good suggestion that it should be placed in the result section. A comment has been added to bring the journal’s attention to this request.

In NCBI BlastN search, it means that query and search sequence share low degree(s) of sequence similarity (No significant similarity found) and results are not displayed. A footnote has been added to Table 1.

Line 144: AspGD is offline. Please refer to and cite fungiDB and/or ENSEMBLfungi instead.

AFLA_009845 AFLA_009846 AFLA_009847

As suggested, the three gene designations have been included in the M&M section. Please also see the reply to the next comment below.

Line 154f: I could not find AFLA_113120 in any database, instead AFLA_007258 should be the accession for gpiA in current releases.

The reason for this discrepancy is due to the recent update to the third version annotation (AAIH03000000, please see Revised Transcriptome-Based Gene Annotation for Aspergillus flavus Strain NRRL 3357.  Microbiology Resource Announcements Vol. 9, No. 49, 3 December 2020). The opinion of a few researchers who are aware of the unexpected consequence is that an updating is reasonable, but the authors should have kept using the same gene designation if no new information about a gene annotation is added. Unfortunately, they did not take into consideration of this concern. For example, the newly known AFLA_007258 is identical to AFLA_113120 in term of gene annotation (same gene size, introns, etc.). Similarly, the three genes of the A. flavus kojic acid gene cluster (kojA, kojR, and kojT) previously known as AFLA_096040, AFLA_096050, and AFLA_096060 are now updated as AFLA_009845, AFLA_009846, and AFLA_009847 as mentioned above by the reviewer. But their gene annotations are still the same. In other words, A. flavus gene designations in publications related to gene function and transcriptomic studies in the past decade have mostly been changed, which has an adverse effect on literature. As commented, the new gene designation has been included in the revised manuscript.

PMID 322111196 should be cited here as well. It describes the use of the gpiA promoter for expression of genes.

This idea for the mentioned 2020 article (PMID 32211196), which surveyed transcriptome datasets for highly expressed genes in Aspergillus oryzae, is same as the cited article- see Table 1 of Aspergillus flavus GPI-anchored protein-encoding ecm33 has a role in growth, development, aflatoxin biosynthesis, and maize infection. Applied Microbiology and Biotechnology volume 102, 5209–5220 (2018). In fact, the A. nidulans gpdA/A. flavus gpiA driven expression data present in the work were generated more than five years (2017) ago. Although the mentioned article is based on the orthologous gene in A. oryzae, as suggested by the review, it has been added to the M&M section in the revised manuscript.

Lines 228ff: It seems that plasmids 191015 and 191016 are no longer available from Addgene?

Because it is now mandatory for USDA to set up Material Transfer Agreement for sharing government-owned research materials, and the process for completing an MTA usually takes six months. Hence, the two plasmids have been deposited at Addgene, a public plasmid repository, for free distribution to the research community (especially after my retirement), which had been approved by USDA Office of Technology Transfer before prior manuscript submission. They are currently being processed by Addgene (quality control) and will soon be made available. Please contact deposit@addgene.org for further information.

Line 270: The current accession numbers of kojA, kojR and kojT should be included.

As commented, the designations of the three gene have been mentioned in the M&M subsection 2.2.

Figure 1: What does "NJ analysis" refer to? Please indicate what is shown by the scale bar. It seems that ATCC12892 is not included in Table 1, why not? It would be good to include the strain names after the species name to exclude any ambiguities.

NJ (Neighbor-Joining) is a bottom-up (agglomerative) clustering method frequently used in bioinformatics for the creation of phylogenetic trees. The scale is the “branch length scale”, which is commonly included in phylogenetic tree without explanation in nearly all bioinformatics articles. Branch lengths indicate genetic change i.e., the longer the branch, the more genetic change (or divergence) has occurred. As commented, ATCC12892 is now included in Table 1. Information about the bar has been added to the figure legend. For a concise introduction about branch length, please click the link below.

https://www.ebi.ac.uk/training/online/courses/introduction-to-phylogenetics/what-is-a-phylogeny/aspects-of-phylogenies/branches/

Table 2: In the footer, footnote "b" might refer to "c" in the table and description of footnote "b" seems to be missing? Please check.

Note for “c” was accidentally omitted when transferring Table 2 to the final version. As commented, it has been included in the revised manuscript.

c: Percentages of amino acid sequence identity to annotated KojR and KojT of P. nordicum DAOMC 185683, which were treated as 100.0, respectively.

Line 328: By reading the manuscript I was unsure about the correct use of "respectively" several times. Here, I am pretty sure that "respectively" should be removed.

As commented, the word has been removed and sentence rephrased.

Table 3: "Control/KUPG" should probably be replaced with "Control (KuPG)". It should be indicated in the footer, what the numbers below "kojA" and "kojT" refer to. Sequence identity? For which region? There also is a "basis region" in one of the column headers.

As commented, KuPG has been used. Now a footnote below also has been included in the revised Table 3.

 a: KuPG is a pyrG+ parental strain of SRRC1709, which was the transformation recipient strain (see M&M).

The other comments are about Table 4. I thank the reviewer for pointing out the typo. The correct word for “basis” should be “basic”. It refers to the basic amino-acid region immediately following the Zn(II)2Cys6 DNA-binding domain. If the review is interested in information about the so-called C6-type zinc-finger regulatory proteins, the article I wrote a decade’s ago “Genome-wide analysis of the Zn(II)2Cys6 zinc cluster-encoding gene family in Aspergillus flavus. Appl Microbiol Biotechnol (2013) 97:4289–4300” may provide some useful information. There are also a few published articles about genome-wide fungal C6-type genes/proteins available. in recent years. The reviewer surely can easily pull them out from PubMed.

The numbers below kojA and kojT refer to the degrees of nucleotide sequence identity of the putative promoter regions compared to those of A. flavus. I apologize for not including supplemental Tables S1 and S2 (only four supplemental figures were submitted) in previous submission. I have added both Tables S1 and S2 to the revised Supplemental Materials. A new footnote under Table 4 has been added as below.

b: See Table S2 for promoter sequences of kojA and kojT. Percentages of nucleotide sequence identity are shown.

Round 2

Reviewer 1 Report

In accordance with the three reviewers’ comments, the revised manuscript has been improved to some extent but requires appropriate revisions to clear the following unsound points in order to be accepted.

1. The authors' response to the reviewer’s comment 1 is the minimum acceptable, but as also noted with the reviewer #3, the expression levels of kojA and kojT should be examined, unless there is a convincing reason why qRT-PCR was not performed here, even though detailed qRT-PCR was done on the overexpression strains. This would be required to improve the scientific quality of the manuscript.

2. The explanation for why the production level of kojic acid by overexpression of kojR is inconsistent with that in the report in A. oryzae is inadequate and unconvincing. However, perhaps one representative of the highly productive strains was selected to present the data in the report, but there was no way to confirm this, so the authors' response might be accepted.

Apart from that, the data in Fig. 2 and Table S3 are quite questionable. In Fig. 2A and 2B, overexpression of kojR reflecting its copy numbers integrated was observed, but curiously the expression of kojA and kojT was not upregulated compared to the control strain in which kojR was expressed at low level with its own promoter. These phenomona should be convincingly explained or discussed.

In addition, the sentences in lines 346 to 361 using the data shown in Table S3 could not be understood. In other words, what the figures shown in Table S3 mean cannot be understood. For example, kojA of D-16 is 12.03 for 48 h and 2.93 for 72 h, but the reviewer has no idea how the fold change is calculated to be 550.6 based on those figures.

3. Fig. 4A and B: The explanation of the photos of the agar plates is still inadequate. The strain of each agar plate should be clearly stated above the photos. It is also incomprehensible why alternatively front and reverse sides were shown, although all of them are enough to include a photo of the reverse sides.

4. The species name “A. awamori” is invalid and this species in the text and Table 2 should be deleted. In addition, two strains of the same species (A. niger, A. tubingensis, A. luchuensis) are unnecessary in Table 2 and one of them should be deleted. Deletion of these should not affect the content of this paper at all.

Author Response

  1. The authors' response to the reviewer’s comment 1 is the minimum acceptable, but as also noted with the reviewer #3, the expression levels of kojA and kojT should be examined, unless there is a convincing reason why qRT-PCR was not performed here, even though detailed qRT-PCR was done on the overexpression strains. This would be required to improve the scientific quality of the manuscript.

In fact, the A. nidulans gpdA/A. flavus gpiA driven expression data present in the work were generated more than five years (2017) ago. Two co-authors have retired since then. I will soon conclude my 35 years’ research career and no longer have technical support in the lab to continue this line of work, although this is a very good suggestion. It still has a long way to go in term s of KA research. The present work is not meant to resolve all aspects of the regulation of KA biosynthesis but to give supportive evidence to the proposition made by Marui et al. more than a decade ago. However, even with further qRT-PCR data some questions will still remain. For example, a reviewer may ask what if the observed correlation in kojA strains (i.e., the six types of kojA mutants lost KA production) is just a coincidence that solely results from a defective kojA promoter and has nothing to do with KojR binding to the putative motif site. Also, as a direct involvement of kojT in KA production is not clearly defined, qRT-PCR data likely will not be satisfactorily interpreted. The KA gene cluster is an incomplete gene cluster. Hopefully, some research groups in the future will be able to identify other KA structural genes that are directly involved in its formation. Including those KA genes in future gene expression work certainly will be much more meaningful. Furthermore, the direct and conclusive evidence of KojR protein-DNA interaction will have to come from the use of ChIP technology, which can either confirm or reject results from the current study. I hope that no matter how small or narrow the scope of this current work is, it would pave the road by providing experimental evidence for other researchers to continue the pursuit.

  1. The explanation for why the production level of kojic acid by overexpression of kojR is inconsistent with that in the report in A. oryzae is inadequate and unconvincing. However, perhaps one representative of the highly productive strains was selected to present the data in the report, but there was no way to confirm this, so the authors' response might be accepted.

I thank the reviewer for the understanding.

Apart from that, the data in Fig. 2 and Table S3 are quite questionable. In Fig. 2A and 2B, overexpression of kojR reflecting its copy numbers integrated was observed, but curiously the expression of kojA and kojT was not upregulated compared to the control strain in which kojR was expressed at low level with its own promoter. These phenomona should be convincingly explained or discussed.

The reviewer probably is not familiar with interpretation of qRT-PCR data. Table S3 actually shows that kojR expression levels for the control strains at 48 h (9.43) and 72 h (8.77) indeed are lower than those of the overexpression strains (for the D set= 5.59 to 7.71 & the I set 3.87 to 5.53 at 48 h; for the D set= 6.67 to 7.86 & the I set 5.45 to 6.22 at 72 h). The qRT-PCR data cannot be read directly as arithmetical numbers as shown; they are exponential numbers (power). To calculate the expression levels, these exponential numbers need to be further converted. In short, a smaller number (when compared to a reference) means the expression level is higher. Not surprisingly, when an exponential number is converted back to an arithmetic number, the actual expression level is even higher. I will explain briefly about the theory and mathematics behind the qRT-PCR approach in the following reply.

In addition, the sentences in lines 346 to 361 using the data shown in Table S3 could not be understood. In other words, what the figures shown in Table S3 mean cannot be understood. For example, kojA of D-16 is 12.03 for 48 h and 2.93 for 72 h, but the reviewer has no idea how the fold change is calculated to be 550.6 based on those figures.

Firstly, a term often mentioned in RT-PCR is the Ct value. Ct stands for the cycle threshold (Ct) of a sample. This value is given by the qPCR machine after the qPCR reaction has been completed. Simply, it is the cycle number (= rounds of amplification) where the fluorescence (signal) generated by the PCR rection is distinguishable from the background noise (the baseline). For example, a Ct value of 5 means it takes 5 PCR amplifications to see the signal. Suppose a sample initially contains 25 (=32) molecules, and after 5 PCR amplifications the signal appears, the final number of the molecule of interest in the sample theoretically will be 210 (=1,024).

Another term is the ∆Ct value, which is the normalized Ct value (to a so-called house-keeping gene, such as 18S, β-tubulin or others) to ensure the amounts of input samples are standardized. Another way to see it is that, after standardization, how many rounds/cycles of amplification are needed for the molecule of interest in each sample to give a detectable signal (The rule apparently is a lesser amount of molecule of interest needs more cycles of amplification to give a detectable signal). Those numbers at 48 h and at 72 h in Table S3 are ∆Ct values. The reviewer may find that in Marui et al. 2011 article, ∆Ct values were used to generate (gene expression) Figure 1 although they did not describe the normalization process used to get the values in the figure.

The third term is the ∆∆Ct value, which is generated by comparing normalized Ct values to a normalized Ct reference (it is treated as 1 or 100%). The ∆∆Ct value is then used for calculating fold-of-change (or relative ratio) using the 2-∆∆Ct formula as shown in Table S3 those calculated values under “Change”. Figures 3A-C also show fold-of-change (or relative ratio).

Now back to Table S3. For the control strain, the kojR ∆Ct value at 48 h is 9.43, which means that it needs to go through 9.43 PCR cycles to give a detectable kojR signal. In contrast, the overexpression strain D-8 at the same timepoint only needs 5.59 PCR cycles to give a detectable kojR signal, which means that in the input samples before qRT-PCR reaction the overexpression stain D-8 contains much more kojR mRNA. The reviewer may ask how much the difference is in the kojR expression levels between D-8 and the control strain at 48 h, and the answer is 14.4-fold more in D-8 as shown in Figure 2B. This value is derived from the 2-∆∆Ct formula. The reviewer may also use the hypothetical scenario that 210 gives a detectable signal mentioned above. The arbitrary value in the control strain then will be 210-9.43 = 20.57 (it takes 9.43 cycles to reach 10 to give a detectable signal) = 1.48, and in the D-8 strain the arbitrary value will be 210-5.59 = 24.41 (it takes 5.59 cycles to reach 10 to give a detectable signal) = 21.26; the ratio of D-8 to the control (21.26/1.48) thus is about 14.4-fold. Similarly, the question raised by the reviewer about the D-16 strain, the fold-of-change (relative ratio) is calculated from the 2-(12.03-2.93), which is 550.6. I hope these concise explanations and examples will help the reviewer understand the meaning behind those numbers.

  1. Fig. 4A and B: The explanation of the photos of the agar plates is still inadequate. The strain of each agar plate should be clearly stated above the photos. It is also incomprehensible why alternatively front and reverse sides were shown, although all of them are enough to include a photo of the reverse sides.

As commented, the photos corresponding to the mutated sequences shown below now have been labelled. To save space and to make the presentation symmetric, the photos therefore are arranged thus. The display showing alternative views allows readers to see front- or reverse-side morphologies (three for each side to be real instead of using only one representative) of the six-type deletion mutants because they are independent isolates.

  1. The species name “A. awamori” is invalid and this species in the text and Table 2 should be deleted. In addition, two strains of the same species (A. niger, A. tubingensis, A. luchuensis) are unnecessary in Table 2 and one of them should be deleted. Deletion of these should not affect the content of this paper at all.

The reviewer must have extensive knowledge about the Aspergillus section Nigri taxonomy. As commented, the invalid species A. awamori has been deleted from the text. The previous included additional A. niger, A. tubingensis, A. luchuensis isolates (the purpose was to show that the KA gene annotations are indeed correct without doubt because genomes of both isolates from the same species when annotated by different research groups have consecutive numbers, which indicates that genes encoding the proteins are situated next to each other) also have been deleted.

Reviewer 3 Report

The authors were able to clarify some of issues raised and have made amendments that clearly improved the manuscript. They also have added a statement indicating that the present study serves as initial characterization of the motif. Therefore, although no experiments could be made and therefore some questions still remain, I suggest publication after minor changes.

Regarding the presented motif, I would suggest to place Fig. 3 after Tab. 5, because as far as I can imagine, the sequences displayed in Tab. 5 were used to generate the logo shown in Fig. 3. When using the promoter sequences provided as Tab S2 I get a different consensus sequence, however, only when restricting the motif width to 11 – CGGCWAAGTCG. At least, authors should explain in detail, how they achieved the presented consensus sequence.

In Fig. 4, the shown colony images should be labeled.

I am sorry that I missed the fact that the CRISPR-Cas9 experiments were performed with a wild type strain.

Author Response

The authors were able to clarify some of issues raised and have made amendments that clearly improved the manuscript. They also have added a statement indicating that the present study serves as initial characterization of the motif. Therefore, although no experiments could be made and therefore some questions still remain, I suggest publication after minor changes.

I thank the reviewer for the recommendation.

Regarding the presented motif, I would suggest to place Fig. 3 after Tab. 5, because as far as I can imagine, the sequences displayed in Tab. 5 were used to generate the logo shown in Fig. 3. When using the promoter sequences provided as Tab S2 I get a different consensus sequence, however, only when restricting the motif width to 11 – CGGCWAAGTCG. At least, authors should explain in detail, how they achieved the presented consensus sequence.

As suggested, Figure 3 has been placed after Table 5. A short description of the criteria use in the MEME analysis also has been included in the figure legend (please see yellow-highlighted text).

“In the MEME analysis, maximum motif width was arbitrarily set at 11 and search for palindromic motifs. Promoter sequences listed in Table S2 are the input sequences.”

In Fig. 4, the shown colony images should be labeled.

As commented, all colonies have been labelled to make them consistent with mutated sequences shown below.

I am sorry that I missed the fact that the CRISPR-Cas9 experiments were performed with a wild type strain.

It’s alright. Readers will now easily understand that the recipient strain used for the CRISPR/Cas9 experiments is a wild-type A. flavus.

Round 3

Reviewer 1 Report

A few unclear and incomplete points still remain in this revised manuscript.

1. By the authors’ response, the reviewer well understood that the figures in Table S3 were ΔCt values and the fold of change was calculated based on those values. However, it seems that such a table showing ΔCt values has been rarely presented in papers published so far. It would be better to redesign the table to be more directly easier to understand for readers.

Since the initial review, the reviewer understood that the fold change value in Fig. 2B was the relative expression level of kojR in overexpression strains compared to that in the control strain, so there is no problem with Fig. 2B. However, because the fold change value in Fig. 2C seems the relative expression level of kojA or kojT in the control and overexpression strains compared to that in the ΔkojR strain, there was no noticeable difference in the kojA and kojT expression levels between the control and overexpression strains at 72 h. What is more curious is that the relative expression levels of kojA and kojT in D-16 and D-20 at 48 h are significantly lower than those in the ΔkojR strain. Thus, the reviewer’s previous comment was posed as to why the expression levels of kojA and kojT in the kojR overexpression strains were not upregulated despite a significant increase in kojR expression levels by its overexpression. This has not been sufficiently explained or discussed.

2. Fig. 4: As the reviewer has previously commented, photos of front sides are inadequite to show the productivity of kojic acid in the promoter mutants, particularly the kojT promoter mutants. For example, photos of front sides of T2, T7, and T24 could not show whether they produced kojic acid because orange color derived from kojic acid could not been observed.

Author Response

  1. By the authors’ response, the reviewer well understood that the figures in Table S3 were ΔCt values and the fold of change was calculated based on those values. However, it seems that such a table showing ΔCt values has been rarely presented in papers published so far. It would be better to redesign the table to be more directly easier to understand for readers.

Thank the reviewer for the comment. The qRT-PCR technology has been around for more than two decades. In other words, it is now a routine technique in research, and presenting “∆Ct raw data” has become a rare practice. The advancement in instrument design and software development over the years have made obtaining qTR-PCR results (calculation of ∆Ct values and relative ratios, that is, 2-∆∆Ct values) an automatic process. This is another reason that the “raw data” have no longer been seen/presented in papers, which is not a surprise. The prepared Table S3 is just for reference purpose; it also is the format frequently seen in the old days. If a reader has a basic understanding of the theory behind qRT-PCT, he or she should be able to quickly grasp the meaning of the data simply by looking at the ∆Ct values and judge increase/decrease in gene expression levels.

Since the initial review, the reviewer understood that the fold change value in Fig. 2B was the relative expression level of kojR in overexpression strains compared to that in the control strain, so there is no problem with Fig. 2B. However, because the fold change value in Fig. 2C seems the relative expression level of kojA or kojT in the control and overexpression strains compared to that in the ΔkojR strain, there was no noticeable difference in the kojA and kojT expression levels between the control and overexpression strains at 72 h. What is more curious is that the relative expression levels of kojA and kojT in D-16 and D-20 at 48 h are significantly lower than those in the ΔkojR strain. Thus, the reviewer’s previous comment was posed as to why the expression levels of kojA and kojT in the kojR overexpression strains were not upregulated despite a significant increase in kojR expression levels by its overexpression. This has not been sufficiently explained or discussed.

Different Aspergillus species/strains have different abilities to produce kojic acid. The production capacity likely is related to the steady-state expression levels of their KA structural genes. As to the reviewer’s first question, there is no definite answer, it can only be speculated that in this A. flavus CA14 strain, the transcription machineries in both the control and the overexpression strains at 72 h have reached/approached their maximum capacity in making the kojA and kojT transcripts (from the basal levels) despite the fact that the kojR transcripts in the overexpression strains indeed were overexpressed as evidenced by all ∆Ct data shown in Table S3. Special and temporal gene expression is intricate and complicate. Regulatory genes such as kojR and structural genes such as kojA and kojT are controlled differently. For example, in the work by Marui et al, the increase in A. oryzae kojR expression at 24 and 72 h was by a factor of 32 (whose expression remained constant over time and is like the finding in this study), but the transcripts of kojA and kojT were barely detectable at 24 h, and their levels at 72 h reached roughly 8- and 3-fold that of the control strain, respectively. Unfortunately, there is an intrinsic drawback in interpreting qRT-PCR data. Researchers are in a dilemma and trained to use gene expression data to infer/correlate results of other gene expression, gene products or related metabolites. In term of deriving a general trend, it is a reasonable practice. But it should not be overlooked that in this case what directly governs synthesis of kojA and kojT transcripts are the transcription machinery plus the KojR regulator and many other unknown (co)factors that operate in a coordinated fashion. It is not known whether all expressed kojR transcripts can be translated mathematically (proportionally) into KojR protein (likely not) or, if yes, can the excess KojR (from overexpression) play a role in increasing the synthesis of kojA or kojT transcripts because of limited supplies other (co)factors? As to the correlation of gene transcripts and proteins, the reviewer may read the past article “Integrative analyses reveal transcriptome-proteome correlation in biological pathways and secondary metabolism clusters in A. flavus in response to temperature. Scientific Report, Bai et al. (2015) 14582”, or a recent article “Integrative analysis of transcriptome and proteome provides insights into adaptation to cadmium stress in Sedum plumbizincicola. Ecotoxicology and Environmental Safety, Zhu et al. (2022) 113149.” which show poor correlation between transcriptome and proteome. Nonetheless, all the explanations do not affect the conclusion that a transcriptional activation mechanism is involved in kojic acid biosynthesis.

As to the reviewer’s question 2, if the reviewer looks Table S3 (larger ∆Ct values mean lower expression levels), the reviewer may get the impression that the expression of kojA and kojT in the heterologous gpdA promoter set strains, D-16 and D-20, somehow were delayed at 48 (I used the word “delayed” because at 72 h their levels were significantly increased.) when compared to those in the homologous gpiA promoter set strains (I-5, I-9, and I-16). The reviewer may also find that D-16 and D-20 contain only a single copy of kojR (Figure 2A). Whether the integration of the gpdA-kojR vector somehow affects the developmental or physiological state of the two strains and results in a delayed (decreased) expression of (basal) kojA and kojT levels at the early period of growth is not known. Any explanations like those described above in nature are purely speculative. Nonetheless, as commented, a short explanation for the observation has been added. Please see “(line 373 in WORD file; line 365 in PDF) ….low at 48 h except for D8…. Noticeably, the kojA and kojT expression levels in D-16 and D-20 were significantly lower than those in the ΔkojR strain. These decreases seem to be related to delayed gene expression at 48 h in the two heterologous gpdA promoter set strains, which contained a single copy of kojR (Figure 2A), as compared to the kojA and kojT expression in the homologous gpiA promoter set strains at 48 h (Table S3). It is not known whether the developmental or physiological state of the two strains has a bearing on the observed variation.

  1. Fig. 4: As the reviewer has previously commented, photos of front sides are inadequite to show the productivity of kojic acid in the promoter mutants, particularly the kojT promoter mutants. For example, photos of front sides of T2, T7, and T24 could not show whether they produced kojic acid because orange color derived from kojic acid could not been observed.

If the reviewer looks closely the photos of T2, T7, and T24 and compare them to those (A3, A7, and A15) of the kojA set, the reviewer will see the distinct color difference around edges of the fungal colonies (front view) between kojic acid producing T strains and non-producing A strains. Because the growth (mycelial mat plus spores) of colonies (front view) masks the underneath orange color (i.e., the KA-iron complex), only those orange complexes diffuse to the colony edge areas can be seen. The reviewer can also look and compare the photos showing the reverse sides of the colonies from the A and T sets (a reason for showing alternative views of the colonies), by this way kojic acid producing T strains can be easily distinguish from those kojic acid non-producing A strains.